# SHAP-IQ: Unified Approximation of any-order Shapley Interactions

**Fabian Fumagalli**[*]
Bielefeld University, CITEC
D-33619, Bielefeld, Germany
`ffumagalli@techfak.uni-bielefeld.de`

**Maximilian Muschalik**[*]
LMU Munich, MCML Munich
D-80539, Munich, Germany
`maximilian.muschalik@ifi.lmu.de`

**Patrick Kolpaczki**
Paderborn University
D-33098, Paderborn, Germany
`patrick.kolpaczki@upb.de`

**Eyke Hüllermeier**
LMU Munich, MCML Munich
D-80539, Munich, Germany
`eyke@ifi.lmu.de`

**Barbara Hammer**
Bielefeld University, CITEC
D-33619, Bielefeld, Germany
`bhammer@techfak.uni-bielefeld.de`

## Abstract

Predominately in explainable artificial intelligence (XAI) research, the Shapley value (SV) is applied to determine feature attributions for any black box model. Shapley interaction indices extend the SV to define any-order feature interactions. Defining a unique Shapley interaction index is an open research question and, so far, three definitions have been proposed, which differ by their choice of axioms. Moreover, each definition requires a specific approximation technique. Here, we propose SHAPley Interaction Quantification (SHAP-IQ), an efficient sampling-based approximator to compute Shapley interactions for arbitrary cardinal interaction indices (CII), i.e. interaction indices that satisfy the linearity, symmetry and dummy axiom. SHAP-IQ is based on a novel representation and, in contrast to existing methods, we provide theoretical guarantees for its approximation quality, as well as estimates for the variance of the point estimates. For the special case of SV, our approach reveals a novel representation of the SV and corresponds to Unbiased KernelSHAP with a greatly simplified calculation. We illustrate the computational efficiency and effectiveness by explaining language, image classification and high-dimensional synthetic models.

## 1 Introduction

Feature attributions are a prevalent approach to interpret black box machine learning (ML) models [2, 7, 26]. However, in many real-world applications, such as understanding drug-drug interactions, mutational events or complex language models, quantifying *interactions* between features is essential, too [48, 23, 43]. Feature interactions provide a more comprehensive explanation, which can be seen as an enrichment of feature attributions [3, 39, 40]. While feature attributions quantify the contribution of *single* features to the model's prediction or performance, feature interactions quantify the contribution of a *group* of features to the model's prediction or performance.

---

[*]denotes equal contribution

37th Conference on Neural Information Processing Systems (NeurIPS 2023).

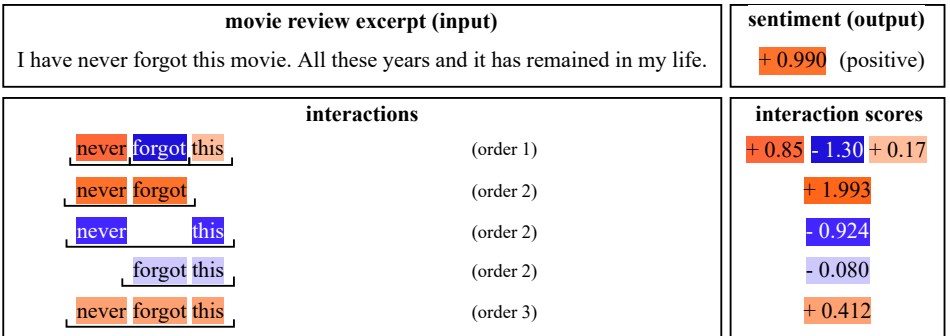

Figure 1: Interaction scores for a movie review excerpt presented to a sentiment analysis model.

In this work, we are interested in feature interactions that make use of the Shapley value (SV) and its extension to Shapley interactions. The SV is a concept from cooperative game theory that has been used, apart from feature attributions [7], as a basis for many Shapley-based explanations [21, 15, 49]. It distinguishes itself through uniqueness given a set of intuitive axioms. A number of approaches extend Shapley-based explanations to feature interactions [17, 3, 39, 40]. Yet, in contrast to the SV, a "natural" extension of the intuitive set of axioms for a unique Shapley interaction index is less clear. Moreover, its efficient computation is challenging and, so far, approximation approaches are specifically tailored to the particular definition.

In this paper, we consider a more general class of interaction indices, known as cardinal interaction indices (CII) [17], which covers all currently proposed definitions and all other that satisfy the (generalized) linearity, symmetry and dummy axiom. We present SHAPley Interaction Quantification (SHAP-IQ), a sampling-based unified approximation method. It is substantiated by mathematical guarantees and can be applied to *any* CII to approximate any-order interaction scores efficiently.

**Contribution.** Our main contributions include:

- We consider a general form of interaction indices, known as CII (Definition 3.3) and establish a novel representation (Theorem 4.1), which we utilize to construct SHAP-IQ (Definition 4.2), an efficient sampling-based estimator.[2]

- We show that SHAP-IQ is unbiased, consistent and provide a general approximation bound (Theorem 4.3). We further prove that SHAP-IQ maintains the efficiency condition for n-Shapley Values [3] and the Shapley Taylor Interaction Index [39] (Theorem 4.7).

- For the SV, we find a novel representation (Theorem 4.4). We further prove that SHAP-IQ is linked to Unbiased KernelSHAP [8] (Theorem 4.5) and greatly simplifies its representation.

- We use SHAP-IQ to compute any-order n-Shapley Values on different ML models and demonstrate that it outperforms existing baseline methods. We further contrast different existing CIIs and compare SHAP-IQ to the corresponding baseline approximation method.

## 2   Related Work

The Shapley Interaction Index (SII) [17], its efficiency preserving aggregation as n-Shapley Values (n-SII) [3], the Shapley Taylor Interaction (STI) [39] and the Faithful Shapley Interaction Index (FSI) [40] offer different ways of extending the SV to interactions, which extend on the linearity, symmetry and dummy axiom to provide a uniquely defined interaction index. SII and STI extend on axiomatic properties of the weighted sum for SV [35], whereas FSI extends on the axiomatic properties of the Shapley interaction as the solution to a weighted least square solution [32, 33].

---

[2]The *shapiq* package extends on the well-known *shap* library and can be found at `https://pypi.org/project/shapiq/`.

In the field of cooperative game theory, interactions have also been studied from a theoretical perspective as the solution of a weighted least square problem and the sum of marginal contributions with constant weights, which both yield a generalized Banzhaf value [18, 16].

In the ML community, interactions of features have been studied from a practical perspective for text [29] and image [41] data, for specific models, such as neural networks [42, 10, 36, 20] or tree based models [25]. Other concepts of interactions have been discussed in [43] using marginal contributions, from a statistical perspective with functional decomposition [28] and improved white box models with interaction terms [24].

To approximate SII and STI, a permutation-based method [39, 40], as an extension of ApproShapley [5], was suggested, whereas FSI relies on a kernel-based approximation, similar to KernelSHAP [26], which utilizes a representation of the SV as the solution of a weighted least square problem [6]. Unlike these specific approaches, we consider general CIIs, which subsume all of the above mentioned measures, and we propose a generic approximation technique, which can be accompanied by mathematical guarantees. In case of the SV, our approximation of the CII is related to Unbiased KernelSHAP [8], which is a variant of KernelSHAP [26]. It is further related to stratified sampling approximations for the SV and our sampling approach can be seen as flexible framework to find the optimum allocation for each stratum [4].

## 3 The Cardinal Interaction Index (CII) and Shapley-based Explanations

In this section, we review Shapley-based explanations and introduce the CII, which we aim to approximate in Section 4. We further introduce existing baseline methods for specific CIIs and Unbiased KernelSHAP for the SV, which is linked to our proposed method.

**Notations.** We refer to the model behavior on a set of features $D = \{1, \ldots, d\}$ as a function $\nu : \mathcal{P}(D) \to \mathbb{R}$, where $\mathcal{P}(D)$ refers to the power set of $D$. We denote $\nu_0(T) := \nu(T) - \nu(\emptyset)$, which is the default setting in game theory and also known as *set function* [17, 14]. The subsets $S \subseteq D$ refer to the set of features (or players in game theory) of which the *interaction* is computed, where we use lower case letters for the cardinality, i.e. $s := |S|$. The maximum order interaction of interest is denoted with $s_0$ and we use the set $\mathcal{T}_k := \{T \subseteq D : k \leq t \leq d - k\}$ and the set of interactions $\mathcal{S}_{s_0} := \{S \subseteq D \mid s \leq s_0, S \neq \emptyset\}$. For a subset $T \subseteq D$, we refer to the binary representation as $Z_T = (z_1, \ldots, z_d) \in \{0, 1\}^d$ with $z_i = \mathbf{1}(i \in T)$ for $i = 1, \ldots, d$, where $\mathbf{1}$ refers to the indicator function. We further denote the Shapley kernel [6, 26] as $\mu(t) := \frac{1}{d-1} \binom{d-2}{t-1}^{-1}$.

Removal-based explanations [9] consider a model that is trained on $d$ features, where the goal is to examine a *model behavior* that is defined on a subset of features. The model behavior $\nu$ for a subset of features could, for instance, be a particular model prediction for one input (local explanation) or an overall measure of model performance (global explanation), if only this subset of features is known [9]. To quantify the contribution for individual features, the change in model behavior is evaluated, if the feature is removed from the model. To restrict a ML model on a subset of features, different *feature removal* techniques have been proposed, such as marginalization of features or retraining the model [9]. To quantify the impact of a single feature $i \in D$ on the model behavior $\nu$ it is then intuitive to compute the difference $\delta^\nu_{\{i\}}(T) = \nu(T \cup \{i\}) - \nu(T)$ for subsets $T \subseteq D \setminus \{i\}$. For a distinct pair of features $(i, j)$ with $i, j \in D$, a natural extension is $\delta^\nu_{\{i,j\}}(T) = \nu(T \cup \{i, j\}) - \nu(T) - \delta^\nu_{\{i\}}(T) - \delta^\nu_{\{j\}}(T)$ for $T \in D \setminus \{i, j\}$, i.e. subtracting the contribution of single features from the joint impact of both features. The following definition generalizes this recursion and is known as *discrete derivative* or *S-derivative* [14].

**Definition 3.1** (Discrete Derivative [14]). *For $S \subseteq D$ the $S$-derivative of $\nu$ at $T \subseteq D \setminus S$ is*

$$\delta^\nu_S(T) := \sum_{L \subseteq S} (-1)^{s-l} \nu(T \cup L).$$

To obtain an attribution score, the marginal contributions on different subsets $T \subseteq D \setminus S$ are aggregated using a specific *summary technique*. In this work, we are interested in the approximation and extension of one particular summary technique for single features $i \in D$, called the Shapley value [35], independent of model behavior and feature removal.

**Definition 3.2** (Shapley Value (SV) [35]). *The SV is $I^{SV}(i) = \sum_{T \subseteq D \setminus \{i\}} \frac{(d-t-1)!t!}{d!} \delta^\nu_{\{i\}}(T)$, $i \in D$.*

The SV is the unique attribution method that fulfills the axioms: symmetry (attributions are independent of feature ordering), linearity (in terms of the model behavior $\nu$), dummy (if a feature does not change $\nu$ then its attribution is zero) and efficiency (the sum of attributions are equal to $\nu_0(D)$) [35]. However, the SV does not give any information about the interactions between two or more features. A suitable extension of the SV for interactions of features $S \subseteq D$ remains an open question, as different axiomatic extensions have been proposed [17, 39, 40]. In this work, we thus consider a broad class of interaction indices, known as CIIs [17, 40], which subsumes popular choices.

**Definition 3.3** (Cardinal Interaction Index (CII) [17]). *A CII is an interaction index of the form*

$$I^m(S) := \sum_{T \subseteq D \setminus S} m_s(t) \delta_S^\nu(T) \text{ with weights } m_s(t) \text{ for } s = 1, \dots, s_0 \text{ and } t = 0, \dots, d - s.$$

**Remark 3.4.** *It was shown that every interaction index satisfying the generalized linearity, symmetry and dummy axioms can be represented as a CII [17]. If $\sum_{t=0}^{d-s} \binom{d-s}{t} m_s(t) = 1$, then the CII is also referred to as a cardinal-probabilistic interaction index (CPII) [14].*

In this paper, we present a unified approximation technique for arbitrary CIIs.

## 3.1 Shapley Interaction Index (SII) and other CIIs

In the following, we introduce prominent examples of CIIs. For further details on the axioms and exact definitions, we refer to the appendix. The SII [17] is a direct extension of the SV, that relies on an additional *recursive* axiom to obtain a unique CII.

**Definition 3.5** (Shapley Interaction Index (SII)[17]). *The SII is a CII defined as*

$$I^{SII}(S) := \sum_{T \subseteq D \setminus S} m_s^{SII}(t) \delta_S^\nu(T) \text{ and } m_s^{SII}(t) := \frac{(d - t - s)! t!}{(d - s + 1)!}.$$

It has been shown that the SII is a CPII [14]. In contrast to the SV, the SII does not fulfill the efficiency axiom, which is a desirable property in the context of ML. Therefore an extension of SII, as well as other interaction indices have been proposed.

**n-Shapley Values (n-SII) and other interaction indices.** The efficiency axiom for interaction indices of *maximum interaction order* $1 \leq s_0 \leq d$ requires that the sum of $I^m(S)$ up to order $s_0$ equals $\nu_0(D)$.

**Definition 3.6** (Efficiency [39, 40]). *A CII is* efficient of order $s_0$, *if $\sum_{S \in \mathcal{S}_{s_0}} I^m(S) = \nu_0(D)$, where $\mathcal{S}_{s_0}$ is the set of interactions up to order $s_0$.*

In [3], an aggregation of SII was proposed to obtain n-SII $I_{s_0}^{\text{n-SII}}(S)$ of order $s_0$ that satisfies efficiency. Other axiomatic approaches directly require efficiency together with the linearity, symmetry and dummy axioms, and omit the recursive axiom of SII. However, in contrast to the SV, this axiom alone does not yield a unique interaction index [39, 40]. The STI [39] requires the efficiency axiom and an additional interaction distribution axiom. On the other hand, the FSI [40] requires the efficiency axiom and the faithfulness property, that relates the interaction index to a solution of a constrained weighted least square problem. The choice of axioms of SII (n-SII), STI and FSI yield a unique interaction index that reduces to the SVs for $s_0 = 1$. For FSI, it was shown that the top-order interactions define a CPII [40, Proposition 21], which is also easily verified for STI. All orders of interactions of FSI and STI can in general be represented as a CII, as they fulfill the linearity, symmetry and dummy axioms [17, Proposition 5]. However, it was noted that for FSI a simple closed-form solution for lower-order interactions in terms of discrete derivatives remains unclear [40, Lemma 70].

## 3.2 Baseline Approximations of SII, STI and FSI.

By definition, the number of evaluations of $\nu$ in $I$, which constitutes the limiting factor in ML, grows exponentially with $d$ and thus, in practice, approximation methods are required. Currently, there does not exist an approximation for the general CII definition, as each index (SII, STI, FSI) requires a specifically tailored technique. Approximations of CII can be distinguished into permutation-based approximation (SII and STI) and kernel-based approximation (FSI). Both extend on existing methods

for the SV, namely permutation sampling [5] for SII and STI, and KernelSHAP [26] for FSI. For a comprehensive overview of the original SV methods, we refer to the appendix. We now briefly discuss existing approaches, which will be used as baselines in our experiments.

**Permutation-based (PB) Approximation for STI and SII [39, 40].**  The permutation-based (PB) approximation computes estimates of SII and STI based on a representation of uniformly sampled random permutations $\pi \sim \text{unif}(\mathfrak{S}_D)$, where $\mathfrak{S}_D$ is the set of all permutations, i.e. the set of all ordered sequences of the elements in $D$. Then,

$$I^{\text{SII}}(S) = \mathbb{E}_{\pi \sim \text{unif}(\mathfrak{S}_D)} \left[ \mathbf{1}(S \in \pi) \delta_S^\nu \left( u_S^-(\pi) \right) \right] \text{ and } I^{\text{STI}}(S) = \mathbb{E}_{\pi \sim \text{unif}(\mathfrak{S}_D)} \left[ \delta_S^\nu \left( u_S^-(\pi) \right) \right].$$

Here, $u_S^-(\pi)$ refers to the set of indices in $\pi$ preceding the first occurrence of any element of $S$ in $\pi$ and $S \in \pi$ is fulfilled, if all elements of $S$ appear as a consecutive sequence in $\pi$. The estimators for SII and STI then compute an approximation by Monte Carlo integration by sampling $\pi \sim \text{unif}(\mathfrak{S}_D)$.

**Kernel-based (KB) Approximation for FSI [40, 8].**  Kernel-based (KB) approximation estimates FSI based on the representation of $I$ as a solution to a constrained weighted least square problem

$$I^{\text{FSI}} = \underset{\beta \in \mathbb{R}^{d_{s_0}}}{\arg \min} \mathbb{E}_{T \sim p(T)}[(\nu(T) - \sum_{\substack{S \in \mathcal{S}_{s_0} \\ S \subseteq T}} \beta(S))^2] \text{ s.t. } \sum_{S \in \mathcal{S}_{s_0}} \beta(S) = \nu(D) \text{ and } \beta(\emptyset) = \nu(\emptyset), \quad (1)$$

where $p(T) \propto \mu(t)$, is a probability distribution over $\mathcal{T}_1$ and $d_{s_0} := |\mathcal{S}_{s_0}|$. KB approximation for FSI estimates the expectation using Monte Carlo integration by sampling from $p(T)$ and solves the approximated least-squares problem explicitly, similar to KernelSHAP [26, 8, 40]. For more details and pseudo code, we refer to the appendix.

### 3.3  Unbiased KernelSHAP (U-KSH) for the SV

U-KSH constitutes a variant of KernelSHAP (KSH) [26], which relies on KB approximation for the SV. In contrast to KSH, U-KSH is theoretically well understood and it was shown that the estimator is unbiased and consistent [8]. U-KSH finds an exact solution to (1) with $s_0 = 1$ as

$$I^{\text{SV}} = A^{-1} \left( b - \mathbf{1} \frac{\mathbf{1}^T A^{-1} b - \nu_0(\mathbf{1})}{\mathbf{1}^T A^{-1} \mathbf{1}} \right) \text{ where } A := \mathbb{E}[ZZ^T], b = \mathbb{E}[Z\nu_0(Z)] \text{ and } p(Z) \propto \mu(t).$$

U-KSH then approximates this solution using Monte Carlo integration.

**Definition 3.7** (Unbiased KernelSHAP (U-KSH) [8])**.** *Given* $T_1, \ldots, T_K \sim p(T) \propto \mu(t)$ *with binary representation* $Z_1, \ldots, Z_K \in \{0,1\}^d$, *U-KSH is defined as*

$$\hat{I}_U^{SV} := A^{-1} \left( \hat{b} - \mathbf{1} \frac{\mathbf{1}^T A^{-1} \hat{b} - \nu_0(\mathbf{1})}{\mathbf{1}^T A^{-1} \mathbf{1}} \right), \text{ where } \hat{b} := \frac{1}{K} \sum_{k=1}^K Z_k \nu_0(Z_k).$$

The main idea of U-KSH is that $A$ can be computed explicitly independent of $\nu$ and only $b$ has to be estimated [8]. By linking U-KSH to our method (Theorem 4.5), we will show that $\hat{I}_U^{\text{SV}}$ can be greatly simplified to a weighted sum.

## 4  SHAP-IQ: Unified Approximation of any-order CII

So far, there exists no unified approximation technique for the general CII. In particular, it is unknown if existing approximation techniques, such PB and KB, generalize to other indices [39, 40, 13]. Furthermore, PB approximation for SII and STI is very inefficient as each update of all estimates requires a significant number of model evaluations. KB approximation for FSI efficiently computes estimates, where one model evaluation can be used to update all interaction scores. It is, however, impossible to compute only a selection of interaction estimates and theoretical results for the estimator are difficult to establish. In the following, we introduce SHAP-IQ (Section 4.1), a unified sampling-based approximation method that can be applied to *any CII*. SHAP-IQ is based on a Monte Carlo estimate of a novel representation of the CII and well-known statistical results are applicable. In the special case of SV, we find a novel representation of the SV and show that SHAP-IQ is linked to

U-KSH (Section 4.2). SHAP-IQ therefore greatly reduces the computational complexity of U-KSH. We further show (Section 4.3), that the sum of interaction estimates of SHAP-IQ remains constant and therefore maintains the efficiency property for STI and SII. Interestingly, for FSI this property does not hold, which should be investigated in future research. All proofs can be found in the appendix.

## 4.1 SHAPley Interaction Quantification (SHAP-IQ)

A key challenge in approximating the CII efficiently is that the sum changes for every interaction subset $S$. We thus first establish a novel representation of the CII. Based on this representation, we construct SHAP-IQ, an efficient estimator of the CII. We show that SHAP-IQ is unbiased, consistent and provide a general approximation bound.

Our novel representation of the CII is defined as a sum over all subsets $T \subseteq D$. In previous works, it was shown that such a representation does exist for games with $\nu(\emptyset) = 0$, if the linearity axiom is fulfilled [17, Proposition 1]. We now explicitly specify this representation and show that the weights, for a CII, only depend on the sizes of $T$ and the intersection $T \cap S$.[3]

**Theorem 4.1.** *It holds* $I^m(S) = \sum_{T \subseteq D} \nu_0(T) \gamma_s^m(t, |T \cap S|)$ *with* $\gamma_s^m(t, k) := (-1)^{s-k} m_s(t - k)$.

Theorem 4.1 yields a novel representation of the CII, where the model evaluations $\nu_0(T)$ are independent of $S$. This allows to utilize every model evaluation to compute all CII scores simultaneously by properly weighting with $\gamma_s^m$. Notably, our representation relies on $\nu_0$ instead of $\nu$, which constitutes an important choice for approximation, on which we elaborate in the appendix.

To approximate $I$, we introduce a *sampling order* $k_0 \geq s_0$, for which we split the sum in Theorem 4.1 to subsets with $T \in \mathcal{T}_{k_0}$ and $T \notin \mathcal{T}_{k_0}$ and rewrite

$$I^m(S) = c_{k_0}(S) + \mathbb{E}_{T \sim p_{k_0}(T)} \left[ \nu_0(T) \frac{\gamma_s^m(t, |T \cap S|)}{p_{k_0}(T)} \right] \text{ with } c_{k_0}(S) := \sum_{T \notin \mathcal{T}_{k_0}} \nu_0(T) \gamma_s^m(t, |T \cap S|),$$

where $p_{k_0}$ is over $\mathcal{T}_{k_0}$. SHAP-IQ then estimates the CII by Monte Carlo integration.

**Definition 4.2** (SHAP-IQ). *The* Shapley Interaction Quantification (SHAP-IQ) of order $k_0$ *with* $K$ *samples is*

$$\hat{I}_{k_0}^m(S) := c_{k_0}(S) + \frac{1}{K} \cdot \sum_{k=1}^{K} \nu_0(T_k) \frac{\gamma_s^m(t_k, |T_k \cap S|)}{p_{k_0}(T_k)} \text{ with } T_1, \ldots, T_K \sim p_{k_0}(T).$$

SHAP-IQ is outlined in the appendix and we establish the following important theoretical guarantees.

**Theorem 4.3.** *SHAP-IQ is unbiased,* $\mathbb{E}\left[\hat{I}_{k_0}^m(S)\right] = I^m(S)$, *and consistent,* $\hat{I}_{k_0}^m(S) \overset{K \to \infty}{\to} I^m(S)$. *With* $\sigma^2(S) := \mathbb{V}\left[\nu_0(T) \frac{\gamma_s^m(|T|, |T \cap S|)}{p_{k_0}(T)}\right]$ *and* $\epsilon > 0$, *it holds* $\mathbb{P}(|\hat{I}_{k_0}^m(S) - I^m(S)| > \epsilon) \leq \frac{1}{K} \frac{\sigma^2(S)}{\epsilon^2}$.

SHAP-IQ provides efficient estimates of all CII scores with important theoretical guarantees. The sample variance $\hat{\sigma}^2$ can further be used for statistical analysis of the estimates.

**Finding the sampling order** $k_0$ **and distribution** $p_{k_0}$. In line with KSH and U-KSH [26, 8], we find $k_0$ in an iterative procedure, outlined in the appendix. We consider *sampling weights* $q(t) \geq 0$ for $0 \leq t \leq d$ that grow symmetrically towards the center and consider a distribution $p_{k_0}(T) \propto q(t)$. Given a budget $M$ and initial $k_0 = 0$, we consider $I^m(S) = \mathbb{E}_{T \sim p_{k_0}(T)}\left[\nu_0(T) \frac{\gamma_s^m(t, |T \cap S|)}{p_{k_0}(T)}\right]$ and iteratively increase $k_0$, if for a subset $T$ of size $k_0$ and $d - k_0$, the condition $M \cdot p_{k_0}(T) \geq 1$ is fulfilled. The budget is then decreased by the number of subsets of that size, i.e. $2\binom{d}{k_0}$. This essentially verifies iteratively, if the expected number of subsets exceeds the total number of subsets. For more details and possible choices of *sampling weights* $q$, we refer to the appendix.

**Computational Complexity.** In contrast to PB approximations, SHAP-IQ allows to iteratively update *all* interaction estimates with *one single* model evaluation for any-order interactions. The

---

[3]Our representation generalizes a result for SII [16, Table 3] to functions with $\nu(\emptyset) \neq 0$ and the class of CIIs.

weights $\gamma_s^m(t, k)$ used for the updates can be efficiently precomputed. The updating process can be implemented efficiently using Welford's algorithm [45], where estimates have to be maintained for all interactions sets, i.e. $d_{s_0}$ in total. In contrast to KB approximation, which requires to solve a weighted least square optimization problem with $d_{s_0}$ variables, the computational effort per interaction increases linearly for SHAP-IQ. Furthermore, SHAP-IQ even allows to update selected interaction estimates, whereas, for instance, KB approximation for FSI requires to estimate all interactions. For more details on the implementation and computational complexity of the baseline methods, we refer to the appendix.

## 4.2 SHAP-IQ for the Shapley Value

In this section, we show that SHAP-IQ, in the special case of single feature subsets $s_0 = 1$, yields novel insights into the SV. Furthermore, SHAP-IQ corresponds to U-KSH and greatly simplifies its calculation. Utilizing Theorem 4.1, we find a novel representation of the SV for every feature $i \in D$.

**Theorem 4.4.** *With $c_1(i) = \frac{\nu_0(D)}{d}$ the SV is $I^{SV}(i) = c_1(i) + \sum_{T \in \mathcal{T}_1} \nu_0(T)\mu(t) \left[ \mathbf{1}(i \in T) - \frac{t}{d} \right]$.*

SHAP-IQ admits a similar form (see appendix) and corresponds to U-KSH $\hat{I}_U^{SV}$.

**Theorem 4.5** (SHAP-IQ simplifies U-KSH). *For $p(T) \propto \mu(t)$ it holds that $\hat{I}_U^{SV} = \hat{I}_1^m$.*

Theorem 4.5 implies that the U-KSH estimator can be computed using the SHAP-IQ estimator, which greatly simplifies the calculation to a weighted sum. The main idea of the proof relies on the observation that not only $A$ can be explicitly computed, but also $A^{-1}$, cf. the appendix.

## 4.3 The Sum of Interaction Scores and SHAP-IQ Efficiency

In this section, we are interested in the sum of CII scores, which we link to a property of SHAP-IQ estimates to maintain the efficiency axiom. By Theorem 4.1, we have $\sum_{S \in \mathcal{S}_{s_0}} I^m(S) = \sum_{T \subseteq D} \nu_0(T) \sum_{S \in \mathcal{S}_{s_0}} \gamma_s^m(t, |T \cap S|)$. For the SV, by Theorem 4.4, this sum is zero for every $T \in \mathcal{T}_1$. For higher order CIIs, we introduce the following definition.

**Definition 4.6.** *A CII is s-efficient, if $\sum_{S \subseteq D, |S|=s_0} \gamma_s^m(t, |T \cap S|) = 0$ for every $T \in \mathcal{T}_{s_0}$.*

**Theorem 4.7.** *SII and STI are s-efficient. In particular, SHAP-IQ estimates maintain efficiency for n-SII and STI.*

Further, if a CII is s-efficient, then the sum of SHAP-IQ estimates remains constant. Although we did not provide a rigorous statement, it is easy to validate numerically that FSI is not s-efficient. This finding suggests that there are conceptional differences between these indices, that should be further investigated in future work. Using s-efficiency it is also possible to find an explicit formula for the sum of interaction scores for SII, which we give in the appendix.

# 5 Experiments

We conduct multiple experiments to illustrate the approximation quality of SHAP-IQ compared to current baseline approaches.[4] We showcase SHAP-IQ estimates on any-order SII (n-SII), on top-order STI and FSI. For each interaction index, we use its specific approximation method as a baseline. For SII and STI, we use the PB approximation and for FSI the KB approximation, further described in the appendix. We then compute n-SII based on the estimated SII values. We compare the baseline methods with SHAP-IQ using $p(T) \propto \mu(t)$. For each iteration we evaluate the approximation quality with different budgets up to a maximum budget of $2^{14}$ model evaluations. To account for variation, we randomly evaluate the approximation method on 50 randomly chosen instances, further described below. To quantify the approximation quality, we compute multiple evaluation metrics for each interaction order: mean-squared error (MSE), MSE for the top-K interactions (MSE@K) and the ratio (precision) of estimated top-K interactions (Prec@K). The top-K interactions are determined in regards to their absolute value.

---

[4]All code and implementations for conducting the experiments can be found at `https://github.com/FFmgll/shapiq`. Running the experiments required a computational cost of approximately 2 000 CPU hours. For more details we refer to the appendix.

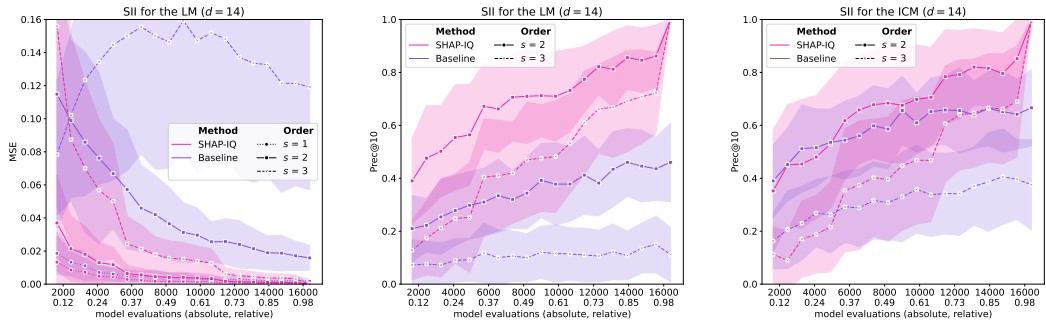

Figure 2: Approximation quality of SHAP-IQ and the baseline for orders $s = 1, 2, 3$ of SII measured by MSE for the LM (left) and Prec@10 for orders $s = 2, 3$ for the LM (middle) and ICM (right).

**Models.** For a language model (LM), we use a fine-tuned version of the DistilBERT transformer architecture [34] on movie review sentences from the original *IMDB* dataset [27, 22] for sentiment analysis, i.e. $\nu$ has values in $[-1, 1]$. In the LM, for a given sentence, different feature coalitions are computed by masking absent features in a tokenized sentence. The implementation is based on the *transformers* API [47]. We randomly sample 50 reviews of length $d = 14$ and explain each model prediction. For an image classification model (ICM), we use ResNet18 [19] pre-trained on ImageNet [11] as provided by *torch* [30]. We randomly sample 50 images and explain the prediction of the corresponding true class. To obtain the prediction of different coalitions, we pre-compute super-pixels with SLIC [1, 44] to obtain a function on $d = 14$ features and apply mean imputation on absent features. For a high-dimensional synthetic model with $d = 30$, we use a *sum of unanimity model* (SOUM) $\nu(T) := \sum_{n=1}^{N} a_n \mathbf{1}(Q_n \subseteq T)$, where $N = 50$ interaction subsets $Q_1, \ldots, Q_N \subseteq D$ are chosen uniformly from all subset sizes and $a_1, \ldots, a_N \in \mathbb{R}$ are generated uniformly $a_n \sim \text{unif}([0, 1])$. Note that the SOUM could also be viewed as an extension of the induced subgraph game [12] for a hypergraph with edges of different order. We randomly generate 50 instances of such SOUMs.

**Ground-Truth (GT) Values.** For the LM and the ICM we compute the ground-truth (GT) values explicitly using the representation from Theorem 4.1. For the high-dimensional SOUM it is impossible to compute the GT values naively. However, due to the linearity of the CII and the simple structure of a SOUM, we can compute the exact GT values of for any CII efficiently, cf. the appendix.

### 5.1 Approximation of any-order SII and n-SII scores using SHAP-IQ

In this experiment, we apply SHAP-IQ on SII and compute estimates for the LM and the ICM up to order $s = 4$. We then compare the estimates with the baseline using the GT values for each order. The results are shown in Figure 2. We display the MSE for the LM (left) and the Prec@10 for the LM (middle) and ICM (right). We further compute the n-SII estimates by aggregating the SII estimates with $s_0 = 4$ and visualize positive and negative interactions on single individuals as proposed in [3]. Thereby, interactions are distributed equally among each participating feature, which was justified in [3, Theorem 6]. This representation amplifies the variance of our sampling-based estimator. We thus also present SHAP-IQ without sampling, i.e. $c_{k_0}$. The results are shown in Figure 3 (left) and from left to right: GT values, SHAP-IQ, SHAP-IQ without sampling and baseline. Lastly, we illustrate the n-SII scores estimates for $s_0 = 3$ of a movie review excerpt classified by the LM (right), where the interactions ("is","not"), ("not","bad"), and ("'ll","love","this") yield a highly positive score.

The results show that SHAP-IQ outperforms the baseline methods across different models and metrics. For the n-SII visualization, we conclude that the SHAP-IQ estimator without sampling is preferable, which yields more accurate results than SHAP-IQ and the baseline methods. In general, SHAP-IQ without sampling performs surprisingly strong, and we encourage further work in this direction.

### 5.2 Approximation of different CIIs using SHAP-IQ

In this experiment, we apply SHAP-IQ on different CIIs, namely SII, STI and FSI. We compute top-order interactions for $s_0 = 3$ and compare the results with the baselines. Our results are shown in Figure 4 (left) and further experiments and results can be found in the appendix. For the LM,

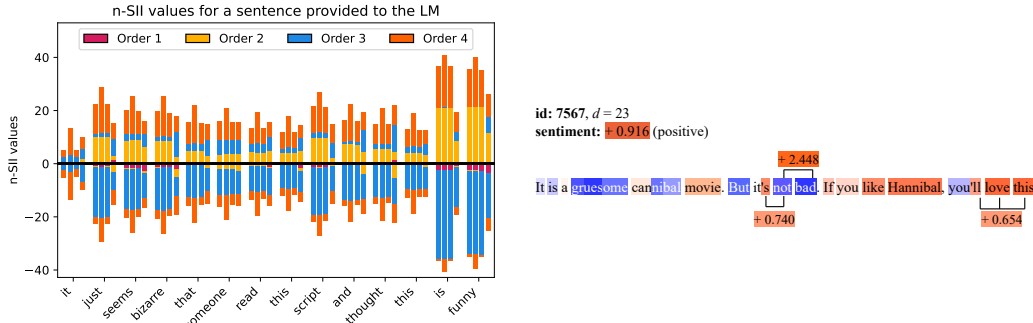

Figure 3: Visualization of n-SII and $s_0 = 4$ [3] (left) with (from left to right): GT, SHAP-IQ, SHAP-IQ without sampling, and baseline. Estimated n-SII scores ($s_0 = 3$) for a movie review (right).

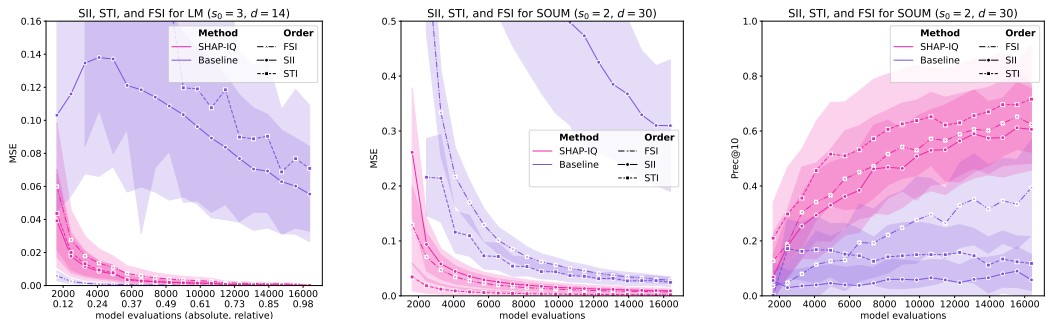

Figure 4: Approximation quality for top-order interactions of SII, STI, and FSI of the LM with $s_0 = 3$ (left) and the SOUM with $s_0 = 2$ (middle and right).

SHAP-IQ clearly outperforms the baseline for SII and STI. For FSI, SHAP-IQ is outperformed by the KB approximation of the baseline. As SII and STI rely on PB approximation, our results indicate that KB approximation is more effective than PB approximation for this setting, which is in line with the strong performance of KernelSHAP [26] for the SV. However, SHAP-IQ, in contrast to KB approximation, provides a solid mathematical foundation with theoretical guarantees and we now consider a high-dimensional synthetic game, where SHAP-IQ outperforms all baselines.

**SHAP-IQ on high-dimensional synthetic models.** For the SOUM we compute the average and standard deviation of each evaluation metric for SHAP-IQ and the baselines for pairwise interactions ($s_0 = 2$) of each index. Our results are shown in Figure 4 (middle and right) and further experiments and results can be found in the appendix. SHAP-IQ outperforms all baseline methods in this setting, in particular, the KB approximation of FSI that performed strongly in ML context. The experiment highlights that there exists no approximation method that performs universally best.

**Runtime Analysis.** The runtime of SHAP-IQ is affected by different parameters. The computation of the sampling order $k_0$ is a constant time operation given a number of features (cf. Algorithm 2 in the appendix). While the pre-computation of the weights ($m_s$) scales linearly with the number of features, the additional computational burden is negligible as it does not depend on $\nu_0$. The main computational cost stems from the model evaluations (access to the value function $\nu_0$), which is bounded by a model's inference time. To illustrate the runtime performance, we compare SHAP-IQ with the baseline methods on the LM using different number of model evaluations $K$. Figure 5 displays the runtime of SHAP-IQ and the corresponding baseline approaches, including all pre-computations. With increasing $K$ the runtime complexity scales linearly, but the overhead of SHAP-IQ remains low. Note that the difference in STI can be attributed to less than $K$ model evaluations, which is required to maintain efficiency, cf. lines 15-16 in Algorithm 6 of the appendix.

## 6    Limitations

We presented SHAP-IQ, a unified approximation algorithm for any-order CIIs with important theoretical guarantees. SHAP-IQ relies on the specific structure in terms of discrete derivatives from Definition 3.3. This representation exists for every interaction indices that fulfills the linearity, symmetry and dummy axiom [17, Proposition 5]. However, for FSI, which has been defined as the solution to the weighted least square problem, a closed-form representation for lower-order interactions in terms of discrete derivatives remains difficult to establish [40, Lemma 70]. This limits the applicability of SHAP-IQ to top-order interactions of FSI, for which this representation is given in [40, Theorem 19]. The FSI baseline performs strongly in ML context, in line with empirical findings for KernelSHAP [26] for the SV. However, the estimator is theoretically not well understood [8] and we have shown that it is not universally best. Moreover, for other CIIs, such as SII and STI, it is unlikely [13] that such an explicit form in terms of a weighted least square problem can be found, which limits the applicability of KB approximation to FSI. SHAP-IQ outperforms the baselines of SII and STI by a large margin, is generally applicable and supported by a solid mathematical foundation.

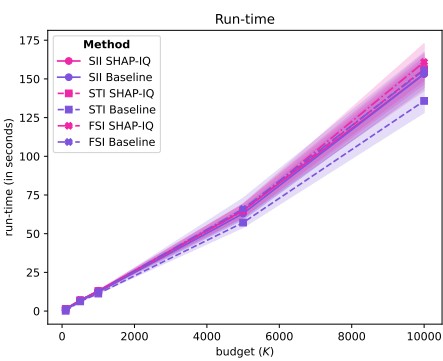

Figure 5: For each interaction index (SII: solid, STI: dashed, FSI: dashdotted), we compare the run-time (in seconds) of SHAP-IQ (pink) compared to baseline estimators (violet) of one instantiation of the LM (16 words) over five independent runs at different levels of K.

## 7    Conclusion

How to extend the SV to interactions is an open research question. In this work, we considered CIIs, a broad class of interaction indices, which covers all currently proposed indices, as well as all indices that fulfill the linearity, symmetry and dummy axiom. We established a novel representation of the CII, which we used to introduce SHAP-IQ, an efficient sampling-based approximation algorithm that is unbiased and consistent. For the special case of SV, SHAP-IQ can be seen as a generalization of U-KSH [8] and greatly simplifies its calculation as well as providing a novel representation of the SV. Furthermore, for n-SII and STI, SHAP-IQ maintains the efficiency condition, which is a direct consequence of a specific property, which we coin s-efficiency for CIIs. We applied SHAP-IQ in multiple experimental settings to compute any-order interactions of SII and n-SII, where SHAP-IQ consistently outperforms the baseline method and showcased the applicability of feature interaction scores to understand black-box language and image classification models. SHAP-IQ further benefits from a solid statistical foundation, which can be leveraged to improve the approximation quality.

**Future work.**    Applying SHAP-IQ to real-world applications, such as NLP tasks [43] and genomics [48, 23], could yield valuable insights. However, the exponentially increasing number of interactions requires human-centered post-processing to enhance interpretability for practitioners and ML engineers, e.g. through automated dialogue systems [37]. Further, it would be beneficial to discover the statistical capabilities of SHAP-IQ to provide confidence bounds or approximate interaction scores sequentially. Beyond model-agnostic approximation, model-specific variants could substantially reduce computational complexity. For instance, it is likely that ideas of TreeSHAP [25] for tree-based models can be extended to Shapley-based interactions.

## Acknowledgements

We sincerely thank the anonymous reviewers for their work and helpful comments. We gratefully acknowledge funding by the Deutsche Forschungsgemeinschaft (DFG, German Research Foundation): TRR 318/1 2021 – 438445824. Patrick Kolpaczki is supported by the research training group Dataninja (Trustworthy AI for Seamless Problem Solving: Next Generation Intelligence Joins Robust Data Analysis) funded by the German federal state of North Rhine-Westphalia.

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

# Appendix of "SHAP-IQ: Unified Approximation of any-order Shapley Interactions"

## Organisation of the Appendix

We provide further theoretical and experimental results for SHAP-IQ. The appendix is organized as follows: In Appendix A, we formally introduce specific CIIs, such as SII, n-SII, STI and FSI and their axiomatic foundation and theoretical results and provide a novel theoretical result for the sum of SII scores. In Appendix B, we provide all proofs of theoretical results from the main paper. In Appendix C, we give further insights into the implementation of SHAP-IQ. In Appendix D, we provide further experimental results, information about the used models, explicit formulas for the SOUM interaction scores, and pseudo-code and information on the computational complexity of baseline implementations. In Appendix E, we provide further theoretical results for the special case of SV, in particular the explicit form of the covariance matrix from [8] and a simplified representation, similar to Theorem 4.4, of SHAP-IQ in this case. In Appendix F, we describe the approximation methods ApproShapley [5] and KernelSHAP [26] for the SV on which our baseline methods are built.

# A  Shapley-based Interaction Indices and further theoretical Results

In this section, we review the axiomatic structures of n-SII, SII, STI and FSI. We further provide an additional result for the sum of SII interaction scores.

## A.1  Shapley-based Interaction Indices

We consider $\mathcal{G}$ as the set of all games $\nu : \mathcal{P}(D) \to \mathbb{R}$. In the following, we formalize the axioms for the Shapley interaction indices $I_\nu : \mathcal{P}(D) \to \mathbb{R}$

**Definition A.1** (Linearity Axiom [17]). *$I_\nu$ is linear, if for any two games $\nu_1, \nu_2 \in \mathcal{G}$ and any $S \subseteq D$, it holds $I_{\nu_1+\nu_2}(S) = I_{\nu_1}(S) + I_{\nu_2}(S)$.*

**Definition A.2** (Dummy Axiom [17]). *For a dummy player $i \in D$ for a game $\nu \in \mathcal{G}$, i.e. constant contribution $c(i)$ added to any coalition $\nu(T \cup \{i\} = \nu(T) + c(i)$, then for every $T \subseteq D \setminus \{i\}$, the dummy axiom requires $I_\nu(S \cup \{i\}) = 0$ for any $S \subseteq D \setminus \{i\}$. That is, a dummy player has no interaction with any coalition.*

**Definition A.3** (Symmetry Axiom [17]). *$I_\nu$ is said to fulfill the symmetry axiom, if for any permutation $\pi$ on $D$ it holds $I_\nu(S) = I_{\pi\nu}(\pi S)$, where $\pi\nu(\pi S) := \nu(S)$ and $\pi S := \{\pi(i) : i \in S\}$ changes the ordering of the players.*

**Definition A.4** (Recursive Axiom [17]). *$I_\nu$ fulfills the recursive axiom, if for any $S \subseteq D$ with $|S| > 1$ and any game $\nu \in \mathcal{G}$*

$$I(S) = I_{\nu_{[S]}}([S]) - \sum_{K \subsetneq S, K \neq \emptyset} I_{\nu^{D \setminus K}}(S \setminus K),$$

*where $\nu_{[S]}$ is the game, where all players in $S$ is considered as one player, and $\nu^{D \setminus K}$ is a game defined on the subset of players $D \setminus K$.*

The recursive axiom defines higher order interactions using lower order interaction. For pairwise interactions it can be stated as $I_\nu(ij) = I_{\nu_{[ij]}}([ij]) - I_{\nu^{D \setminus \{j\}}}(i) - I_{\nu^{D \setminus \{i\}}}(j)$, i.e. the pairwise interaction is the difference of the value for the reduced player $[ij]$ and the individual player values for the reduced game.

**Definition A.5** (Shapley Interaction Index [17]). *The Shapley interaction index (SII) is the unique interaction index that satisfies the linearity, dummy, symmetry and recursive axiom, where the values for $|S| = 1$ correspond to the Shapley value. It can be represented as a CII as*

$$I^{SII}(S) := \sum_{T \subseteq D \setminus S} m_s^{SII}(t) \delta_S^\nu(T) \text{ and } m_s^{SII}(t) := \frac{(d - t - s)! t!}{(d - s + 1)!}.$$

In contrast to the SV, the SII does not yield an efficiency property, which is desirable in ML context. The efficiency axiom was therefore introduced for interactions. The following axioms rely on a maximum interaction order $s_0$, where the values of the interaction indices change for different maximum interaction orders.

**Definition A.6** (Efficiency Axiom [39]). *For all $\nu \in \mathcal{G}$, it holds $\sum_{S \in \mathcal{S}_{s_0}} I_\nu(S) = \nu(D) - \nu(\emptyset)$.*

The efficiency axiom is an extension of the SV efficiency axiom and requires that all interaction scores up to the maximum order $s_0$ sum up to $\nu(D) - \nu(\emptyset)$. For SII there exists a unique recursive aggregation, such that the efficiency axiom is fulfilled. This novel interaction index is referred to as n-Shapley Values (n-SII) [3].

**Definition A.7** (n-Shapley Values (n-SII) [3]). *Given $s_0$ the n-Shapley Values (n-SII) are defined as*

$$I_{s_0}^{n\text{-}SII}(S) := \begin{cases} I^{SII}(S), & \text{for } |S| = s_0 \\ I_{s_0-1}^{n\text{-}SII}(S) + B_{d-|S|} \sum_{\substack{K \subseteq D \setminus S \\ k+s=d}} I^{SII}(S \cup K), & \text{for } |S| < s_0. \end{cases}$$

Besides n-SII, there have been two axiomatic extensions to CIIs that directly require the efficiency axiom together with the linearity, symmetry and dummy axiom. However, unlike for the SV, it is not sufficient to require efficiency for a unique interaction index. The Shapley Taylor Interaction Index (STI) further specifies the interaction distribution axiom, which then yields a unique index [39].

**Definition A.8** (Interaction Distribution Axiom [39]). *For an interaction function $\nu_T$ parametrized by $T \subseteq D$ it holds $\nu_T(S) = 0$, if $T \subsetneq S$ and a constant value $\nu_T(S) = c$ for $T \subseteq S$ and $c \in \mathbb{R}$. The interaction distribution axiom requires for all $S \subseteq D$ with $S \subsetneq T$ and $s < s_0$ that $I_{\nu_T}(S) = 0$.*

**Definition A.9** (The Shapley Taylor Interaction Index [39]). *The Shapley Taylor interaction index (STI) is the unique interaction index that satisfies the linearity, dummy, symmetry, efficiency and interaction distribution axiom.*

The STI yields a unique interaction index by introducing the interaction distribution axiom, which favors the maximum order interactions as discussed in [40]. It was thus argued that instead the representation of the interaction index as a solution to a weighted least square problem is preferable [40], which yields the Faith-Interaction Index.

**Definition A.10** (Faith-Interaction Index [40]). *I is called a* Faith-interaction index *if it can be expressed as*

$$I = \underset{\beta \in \mathbb{R}^{d_{s_0}}}{\arg\min} \sum_{T \subseteq D : \mu(T) < \infty} \mu(T) \left( \nu(T) - \sum_{\substack{T \subseteq S \\ t \leq s_0}} \beta(S) \right)^2 \tag{2}$$
$$s.t. \ \nu(T) = \sum_{S \subseteq T, t \leq s_0} \beta(S), \forall T : \mu(T) = \infty.$$

**Definition A.11** (Faithful Shapley Interaction Index [40]). *The Faithful Shapley interaction index (FSI) is the unique faith-interaction index that satisfies the linearity, dummy, symmetry and efficiency axiom.*

While n-SII, SII, STI and FSI offer differnt ways of characterizing an interaction index, it was shown in [17] that every interaction index satisfying the linearity, symmetry and dummy axiom admits a CII representation.

**Proposition A.12.** *Every interaction index satisfying the linearity, symmetry and dummy axiom can be represented as a CII*

$$I^m(S) := \sum_{T \subseteq D \setminus S} m_s(t) \delta_S^\nu(T).$$

*Furthermore, the weights $m_{s_0}$ for the top-order interaction indices, i.e. $s = s_0$ are defined as*

$$m_{s_0}^{SII}(t) := \frac{(d - t - s_0)! t!}{(d - s_0 + 1)!},$$
$$m_{s_0}^{STI}(t) := s_0 \frac{(d - t - 1)! t!}{d!}$$
$$m_{s_0}^{FII}(t) := \frac{(2s_0 - 1)!}{((s_0 - 1)!)^2} \frac{(d - t - 1)!(t + s_0 - 1)!}{(d + s_0 - 1)!}.$$

*The definitions for lower order interactions can be found in [17] for SII, in [39] for STI and in [40] for FSI. Note that for FSI a closed-form solution for lower-order interactions in terms of discrete derivatives remains unknown [40, Lemma 70].*

*Proof.* For the proofs, we refer to the corresponding paper of each index. The general statement is proven in [17, Proposition 5]. □

## A.2 Explicit Formula for Sum of SII Scores

Using s-efficiency, it is easy to calculate the sum of SII scores, which provide an explicit representation of a formula considered in [31, Theorem 4.2].

**Proposition A.13** (Sum of SII Scores). *For SII it holds*

$$\sum_{\substack{S \subseteq D \\ |S| = s_0}} I^{SII}(S) = \sum_{\substack{T \subseteq D \\ t < s_0}} (-1)^t r(t) \left[ (-1)^{s_0} \nu(T) + \nu(D \setminus T) \right],$$

where $r(t) := \frac{1}{s_0}\binom{d-t}{s_0-t-1}$ and $m_s^{\mathrm{SII}}(t) := \frac{(d-t-s_0)!t!}{(d-s_0+1)!}$.

*Proof.* For SII, we let $m(t) := m_s^{\mathrm{SII}}(t) = \frac{1}{d-s+1}\binom{d-s}{t}^{-1}$ and have by Theorem 4.7 and the definition of s-efficiency

$$\sum_{\substack{S\subseteq D \\ |S|=s_0}} I^m(S) = \sum_{\substack{S\subseteq D \\ |S|=s_0}} c_{s_0}(S) = \sum_{\substack{T\subseteq D \\ t<s_0}} \nu(T) \sum_{\substack{S\subseteq D \\ |S|=s_0}} \gamma_s^m(t,|T\cap S|) + \sum_{\substack{T\subseteq D \\ t>d-s_0}} \nu(T) \sum_{\substack{S\subseteq D \\ |S|=s_0}} \gamma_s^m(t,|T\cap S|)$$

For $t < s_0$ we have

$$\rho(t) := \sum_{\substack{S\subseteq D \\ |S|=s_0}} \gamma_s^m(t,|T\cap S|) = \sum_{k=0}^{t} \binom{t}{k}\binom{t-k}{s_0-k}\gamma_s^m(t,|T\cap S|)$$

$$= \frac{1}{d-s_0+1}\sum_{k=0}^{t}(-1)^{s_0-k}\binom{t}{k}\binom{d-t}{s_0-k}\binom{d-s_0}{t-k}^{-1}.$$

For $t > d-s$ there are at least $k_{\min} = t-(d-s)$ elements in the intersection of $|T\cap S|$ and thus with $\bar{t} := d-t < s_0$ and $k_{\min} = s_0 - \bar{t}$

$$\sum_{\substack{S\subseteq D \\ |S|=s_0}} \gamma_s^m(t,|T\cap S|) = \sum_{k=k_{\min}}^{s_0} \binom{t}{k}\binom{t-k}{s_0-k}\gamma_s^m(t,|T\cap S|)$$

$$= \frac{1}{d-s_0+1}\sum_{k=k_{\min}}^{s_0}(-1)^{s_0-k}\binom{t}{k}\binom{d-t}{s_0-k}\binom{d-s_0}{t-k}^{-1}$$

$$= \frac{1}{d-s_0+1}\sum_{k=s_0-\bar{t}}^{s_0}(-1)^{s_0-k}\binom{d-\bar{t}}{k}\binom{\bar{t}}{s_0-k}\binom{d-s_0}{d-\bar{t}-k}^{-1}$$

$$= \frac{1}{d-s_0+1}\sum_{k=0}^{\bar{t}}(-1)^{k}\binom{d-\bar{t}}{s_0-k}\binom{\bar{t}}{k}\binom{d-s_0}{d-\bar{t}-s_0+k}^{-1}$$

$$= \frac{1}{d-s_0+1}\sum_{k=0}^{\bar{t}}(-1)^{k}\binom{d-\bar{t}}{s_0-k}\binom{\bar{t}}{k}\binom{d-s_0}{\bar{t}-k}^{-1}$$

$$= (-1)^{s_0}\rho(\bar{t}).$$

We can explicitly compute $\rho(t)$ as

$$\rho(t) = \frac{1}{d-s_0+1}\sum_{k=0}^{t}(-1)^{s_0-k}\binom{t}{k}\binom{d-t}{s_0-k}\binom{d-s_0}{t-k}^{-1}$$

$$= (-1)^{s_0}\frac{t!(d-t)!}{(d-s_0+1)!}\sum_{k=0}^{t}(-1)^{k}\frac{1}{(s_0-k)!k!}$$

$$= (-1)^{s_0}\frac{t!(d-t)!}{s_0!(d-s_0+1)!}\sum_{k=0}^{t}(-1)^{k}\binom{s_0}{k}$$

$$= (-1)^{s_0}\frac{t!(d-t)!}{s_0!(d-s_0+1)!}(-1)^{t}\binom{s_0-1}{t} = \frac{(-1)^{s_0+t}}{s_0}\binom{d-t}{s_0-t-1},$$

where we have used that $\sum_{k=0}^{t}(-1)^{k}\binom{s_0}{k} = (-1)^{t}\binom{s_0-1}{t}$ for $t < s_0$. Hence,

$$\sum_{\substack{S\subseteq D \\ |S|=s_0}} I^m(S) = \frac{1}{s_0}\sum_{\substack{T\subseteq D \\ t<s_0}} [(-1)^{s_0}\nu(T) + \nu(D\setminus T)](-1)^{t}\binom{d-t}{s_0-t-1}.$$

$\square$

# B Proofs

This section contains the proofs of the claims made in the main paper.

## B.1 Proof of Theorem 4.1

*Proof.* By definition, the sum $I^m(S) := \sum_{T \subseteq D \setminus S} m(t) \delta_S^\nu(T)$ ranges over all subsets $T \subseteq D$, where every subset is exactly once evaluated. On the one hand, it is easy to see that every evaluated subset in $I^m(S)$ is different, as $T \cup L$ is unique. Furthermore, given any subset $T \subseteq D$, we decompose $T = \tilde{T} \cup L$, where $\tilde{T} \subseteq D \setminus S$ and $L := T \cap S \subseteq S$. The corresponding weight is $m(\tilde{t}) = m(t - l) = m(t - |T \cap S|)$ and the sign from $\delta_S^\nu(\tilde{T})$ is $(-1)^{s-l} = (-1)^{s-|T \cap S|}$. This yields with $\nu_0(T) := \nu(T) - \nu(\emptyset)$

$$
\begin{aligned}
I^m(S) &= \sum_{T \subseteq D} \nu(T) \gamma_s^m(t, |T \cap S|) = \sum_{T \subseteq D} \nu_0(T) \gamma_s^m(t, |T \cap S|) + \nu(\emptyset) \sum_{T \subseteq D} \gamma_s^m(t, |T \cap S|) \\
&= \sum_{T \subseteq D} \nu_0(T) \gamma_s^m(t, |T \cap S|),
\end{aligned}
$$

as the sum over all $\gamma^m$ is zero, if the dummy axiom is fulfilled. $\qquad\square$

**Remark B.1.** *It is important to note, that the sum $\sum_{T \subseteq D} \gamma_s^m(t, |T \cap S|)$ is not zero, if not all subsets are considered, which makes it crucial to use $\nu_0$ instead of $\nu$. In fact, the estimates of $I^m$ would be heavily skewed by $\nu(\emptyset)$. While the estimator would still be unbiased, its variance would scale with $\nu(\emptyset)^2$.*

## B.2 Proof of Theorem 4.3

*Proof.* We aim to show that $\hat{I}^m(S)$ is unbiased and consistent, i.e. $\mathbb{E}\left[\hat{I}_{k_0}^m(S)\right] = I^m(S)$ and $\lim_{K \to \infty} \hat{I}_{k_0}^m(S) = I^m(S)$. Given

$$
\hat{I}_{k_0}^m(S) := c_{k_0}(S) + \frac{1}{K} \cdot \sum_{k=1}^K \nu_0(T_k) \frac{\gamma_s^m(t_k, |T_k \cap S|)}{p_{k_0}(T_k)},
$$

it is clear that due to the linearity of the expectation

$$
\mathbb{E}_{T \sim p_{k_0}(T)}\left[\hat{I}_{k_0}^m(S)\right] = c_{k_0}(S) + \frac{1}{K} \sum_{k=1}^K \mathbb{E}_{T \sim p_{k_0}(T)}\left[\nu_0(T) \frac{\gamma_s^m(|T|, |T \cap S|)}{p_{k_0}(T)}\right] = I^m(S).
$$

Furthermore, let $\sigma^2(S) := \mathbb{V}_{T \sim p_{k_0}(T)}\left[\nu_0(T) \frac{\gamma_s^m(|T|, |T \cap S|)}{p_{k_0}(T)}\right]$ be the variance of each estimate, then, by the law of large numbers

$$
\frac{1}{K} \cdot \sum_{k=1}^K \nu_0(T_k) \frac{\gamma_s^m(t_k, |T_k \cap S|)}{p_{k_0}(T_k)} \xrightarrow{K \to \infty} \mathbb{E}_{T \sim p_{k_0}(T)}\left[\nu_0(T) \frac{\gamma_s^m(|T|, |T \cap S|)}{p_{k_0}(T)}\right],
$$

and thus $\lim_{K \to \infty} \hat{I}^m(S) = I^m(S)$. Lastly, as $\hat{I}^m(S)$ is unbiased, we have for $\epsilon > 0$ by Chebyshev's inequality

$$
\mathbb{P}(|\hat{I}_{k_0}^m(S) - I^m(S)| > \epsilon) \leq \frac{\mathbb{V}\left[\hat{I}_{k_0}^m(S)\right]}{\epsilon^2} = \frac{1}{K^2} \frac{K \sigma^2(S)}{\epsilon^2} = \frac{1}{K} \frac{\sigma^2(S)}{\epsilon^2}.
$$

$\qquad\square$

## B.3 Proof of Theorem 4.4

*Proof.* We let $m(t) := \frac{(d-t-1)!t!}{d!}$ and apply Theorem 4.1. With $\gamma_s^m(0,0) = -m(0)$, $\gamma_s^m(d,1) = m(d-1)$ and $m(0) = m(d-1) = \frac{1}{d}$, we have

$$I^{\mathrm{SV}}(i) \overset{\text{Theorem 4.1}}{=} \sum_{T \subseteq D} \nu(T)\gamma_s^m(t, \mathbf{1}(i \in T)) = \frac{\nu(D) - \nu(\emptyset)}{d} + \sum_{T \in \mathcal{T}_1} \nu(T)\gamma_s^m(t, \mathbf{1}(i \in T))$$

$$= c_1(i) + \sum_{T \in \mathcal{T}_1} \nu(T) \left[ \mathbf{1}(i \in T)\gamma_s^m(t,1) + \mathbf{1}(i \notin T)\gamma_s^m(t,0) \right]$$

$$= c_1(i) + \sum_{T \in \mathcal{T}_1} \nu(T) \left[ \mathbf{1}(i \in T)\left(\gamma_s^m(t,1) - \gamma_s^m(t,0)\right) + \gamma_s^m(t,0) \right]$$

$$= c_1(i) + \sum_{T \in \mathcal{T}_1} \nu(T) \left[ \mathbf{1}(i \in T)\left(m(t-1) + m(t)\right) - m(t) \right]$$

$$= c_1(i) + \sum_{T \in \mathcal{T}_1} \nu(T) \left[ \mathbf{1}(i \in T)\left( \frac{(d-t)!(t-1)!}{d!} + \frac{(d-t-1)!t!}{d!} \right) - \frac{(d-t-1)!t!}{d!} \right]$$

$$= c_1(i) + \sum_{T \in \mathcal{T}_1} \nu(T) \frac{(d-t-1)!(t-1)!}{(d-1)!} \left[ \mathbf{1}(i \in T) - \frac{t}{d} \right]$$

$$= c_1(i) + \sum_{T \in \mathcal{T}_1} \nu(T)\mu(t) \left[ \mathbf{1}(i \in T) - \frac{t}{d} \right].$$

$\square$

## B.4 Proof of Theorem 4.5

*Proof.* According to Proposition E.3, our goal is to show that

$$\hat{I}_U^{\mathrm{SV}}(i) = c_1(i) + \frac{2h_{d-1}}{K} \sum_{k=1}^{K} \nu_0(T_k) \left[ \mathbf{1}(i \in T_k) - \frac{t_k}{d} \right]$$

with $T_k \overset{\text{iid}}{\sim} p(T) := \mu(t)/(2h_{d-1})$, where $p$ is a probability distribution over $\mathcal{T}_1$ and $h_n := \sum_{t=1}^{n} t^{-1}$. The proof is structured in the following steps:

1. Exact computation of $A^{-1}$ using the exact structure of $A$ with diagonal entries $\mu_1$ and off-diagonal entries $\mu_2$, cf. [8, Appendix A].

2. Exact computation of $\hat{I}_U^{\mathrm{SV}}$, which yields with Proposition E.3 $\hat{I}_U^{\mathrm{SV}}(i) = \hat{I}_1^m(i)$, if $(\mu_1 - \mu_2)2h_{d-1} = 1$.

3. We show that $(\mu_1 - \mu_2)2h_{d-1} = 1$.

**Calculation of $A^{-1}$.** It has been shown in [8, Appendix A] that all off-diagonal entries are equal and all diagonal entries are equal, i.e. $A$ may be written as $A = \mu_2 \mathbf{J} + (\mu_1 - \mu_2)\mathbf{I}$ with off-diagonal entries $\mu_2 := p(Z_i = Z_j = 1)$ and diagonal entries $\mu_1 := p(Z_i = 1)$, where $Z_i$ refers to the i-th component of the binary vector $Z$ and $\mathbf{J}$ is a matrix of ones and $\mathbf{I}$ is the identity matrix. The simple structure of A allows to compute the inverse exactly by using the following Lemma.

**Lemma B.2.** *Let $\mu_1, \mu_2 > 0$ with $\mu_1 \neq \mu_2$, then*

$$(\mu_2 \mathbf{J} + (\mu_1 - \mu_2)\mathbf{I})^{-1} = \tilde{\mu}_2 \mathbf{J} + (\tilde{\mu}_1 - \tilde{\mu}_2)\mathbf{I}$$

*with*

$$\tilde{\mu}_2 = \frac{-\mu_2}{(\mu_1 - \mu_2)(\mu_1 + (d-1)\mu_2)}$$

$$\tilde{\mu}_1 = \frac{\mu_1 + (d-2)\mu_2}{(\mu_1 - \mu_2)(\mu_1 + (d-1)\mu_2)}.$$

*Proof.* We compute

$$\begin{aligned}
\mathbf{I} &= (\mu_2 \mathbf{J} + (\mu_1 - \mu_2)\mathbf{I}) \cdot (\tilde{\mu}_2 \mathbf{J} + (\tilde{\mu}_1 - \tilde{\mu}_2)\mathbf{I}) \\
&= ((\mu_1 + (d-1)\mu_2)\tilde{\mu}_2 + (\tilde{\mu}_1 - \tilde{\mu}_2)\mu_2)\mathbf{J} \\
&\quad + (\mu_1 - \mu_2)(\tilde{\mu}_1 - \tilde{\mu}_2)\mathbf{I},
\end{aligned}$$

which yields $(\mu_1 - \mu_2)(\tilde{\mu}_1 - \tilde{\mu}_2) = 1$ and $\mu_2 \tilde{\mu}_2 d + (\mu_1 - \mu_2)\tilde{\mu}_1 + (\tilde{\mu}_1 - \tilde{\mu}_2)\mu_2 = 0$. From the first equation we have $\tilde{\mu}_1 - \tilde{\mu}_2 = 1/(\mu_1 - \mu_2)$ and thus by the second equation

$$\tilde{\mu}_2 = \frac{-\mu_2}{(\mu_1 - \mu_2)(\mu_1 + (d-1)\mu_2)}$$

and hence

$$\tilde{\mu}_1 = \frac{\mu_1 + (d-2)\mu_2}{(\mu_1 - \mu_2)(\mu_1 + (d-1)\mu_2)}$$

$\square$

**Calculation of $\hat{I}_U^{\text{SV}}$.** By Lemma B.2, we proceed to compute the different components of

$$\hat{I}_U^{\text{SV}} := A^{-1}\left(\hat{b}_L - \mathbf{1}\frac{\mathbf{1}^T A^{-1}\hat{b}_L - \nu_0(\mathbf{1})}{\mathbf{1}^T A^{-1}\mathbf{1}}\right).$$

First,

$$\mathbf{1}^T A^{-1} = ((d-1)\tilde{\mu}_2 + \tilde{\mu}_1)\,\mathbf{1}^T = \frac{1}{\mu_1 + (d-1)\mu_2}\mathbf{1}^T.$$

Then the denominator yields $\mathbf{1}^T A^{-1}\mathbf{1} = \frac{d}{\mu_1 + (d-1)\mu_2}$. We then obtain

$$\mathbf{1}\frac{\mathbf{1}^T A^{-1}\hat{b}_L - \nu_0(\mathbf{1})}{\mathbf{1}^T A^{-1}\mathbf{1}} = \frac{1}{d}\mathbf{1}\mathbf{1}^T \hat{b}_L - \frac{\mu_1 + (d-1)\mu_2}{d}\nu_0(\mathbf{1})\mathbf{1} = \frac{1}{d}\mathbf{J}\cdot\hat{b}_L + \frac{\mu_1 + (d-1)\mu_2}{d}\nu_0(\mathbf{1})\cdot\mathbf{1},$$

which, with $A^{-1}\mathbf{1} = (\tilde{\mu}_1 + (d-1)\tilde{\mu}_2)\mathbf{1} = \frac{1}{\mu_1 + (d-1)\mu_2}\mathbf{1}$, yields

$$\hat{I}_U^{\text{SV}} = A^{-1}(\hat{b}_L - \frac{1}{d}\mathbf{J}\cdot\hat{b}_L) + \frac{\nu_0(\mathbf{1})}{d}\cdot\mathbf{1} = c_1 + A^{-1}(\hat{b}_L - \frac{1}{d}\mathbf{J}\cdot\hat{b}_L).$$

It remains to show that

$$\left(A^{-1}(\hat{b}_L - \frac{1}{d}\mathbf{J}\cdot\hat{b}_L)\right)_i = \frac{1}{K}\sum_{k=1}^{K}\nu_0(T_k)\frac{\gamma_s^m(t_k, \mathbf{1}(i \in T_k))}{p(T_k)}.$$

With $\hat{b}_L = \frac{1}{K}\sum_{k=1}^{K}z_k\nu_0(z_k)$ it follows

$$A^{-1}(\hat{b}_L - \frac{1}{d}\mathbf{J}\cdot\hat{b}_L) = \frac{1}{K}\sum_{k=1}^{K}\left(A^{-1}z_k - \frac{1}{d}A^{-1}\mathbf{J}z_k\right)\nu_0(z_k).$$

Then $A^{-1}z_k = t_k\tilde{\mu}_2\mathbf{1} + (\tilde{\mu}_1 - \tilde{\mu}_2)z_k$ where $t_k$ is the subset size, i.e. $t_k$ is the sum of all entries in $z_k$. It follows with $A^{-1}J = \frac{1}{\mu_1 + (d-1)\mu_2}J$

$$A^{-1}z_k - \frac{1}{d}A^{-1}\mathbf{J}z_k = t_k\left(\tilde{\mu}_2 - \frac{1}{d(\mu_1 + (d-1)\mu_2)}\right)\mathbf{1} + (\tilde{\mu}_1 - \tilde{\mu}_2)z_k = \frac{1}{\mu_1 - \mu_2}\left(z_k - \frac{t_k}{d}\mathbf{1}\right).$$

For the $i$-th component, we have with set notation $T_k$ and for the SV weights $m(t) := \frac{(d-t-1)!t!}{d!}$ then

$$\begin{aligned}
\hat{I}_U^{\text{SV}}(i) - c_1(i) &= \left(A^{-1}(\hat{b}_L - \frac{1}{d}\mathbf{J}\cdot\hat{b}_L)\right)_i = \frac{1}{K}\sum_{k=1}^{K}\nu_0(T_k)\frac{1}{\mu_1 - \mu_2}\left(\mathbf{1}(i \in T_k) - \frac{t_k}{d}\right) \\
&= \frac{1}{(\mu_1 - \mu_2)2h_{d-1}}\left(\hat{I}_1^m(i) - c_1(i)\right),
\end{aligned}$$

where we have used Proposition E.3 for $\hat{I}_1^m(i)$. It remains to show that $(\mu_1 - \mu_2)2h_{d-1} = 1$.

**Show that** $(\mu_1 - \mu_2)2h_{d-1} = 1$. We let $p(T_k) := \mu(t)/R$ be a probability distribution over $\mathcal{T}_1$. By definition, and as subsets of size $t$ have equal probability, we have

$$\mu_1 = p(Z_i = 1) = \sum_{t=1}^{d-1} p(Z_i = 1 | \mathbf{1}^T Z = t) p(\mathbf{1}^T Z = t)$$

$$= \sum_{t=1}^{d-1} \frac{\binom{d-1}{t-1}}{\binom{d}{t}} \frac{\mu(t)}{R} \binom{d}{t} = \sum_{t=1}^{d-1} \binom{d-1}{t-1} \frac{\mu(t)}{R}$$

and

$$\mu_2 = p(Z_i = Z_j = 1) = \sum_{t=1}^{d-1} p(Z_i = Z_j = 1 | \mathbf{1}^T Z = t) p(\mathbf{1}^T Z = t)$$

$$= \sum_{t=2}^{d-1} \frac{\binom{d-2}{t-2}}{\binom{d}{t}} \frac{\mu(t)}{R} \binom{d}{t} = \sum_{t=2}^{d-1} \binom{d-2}{t-2} \frac{\mu(t)}{R}.$$

Hence,

$$\mu_1 - \mu_2 = \frac{\mu(1)}{R} + \sum_{t=2}^{d-1} \frac{\mu(t)}{R} \left( \binom{d-1}{t-1} - \binom{d-2}{t-2} \right) = \sum_{t=1}^{d-1} \frac{\mu(t)}{R} \binom{d-2}{t-1} = \frac{1}{R},$$

where we have used the recursion for the binomial coefficient and $\mu(t) = \frac{1}{d-1} \binom{d-2}{t-1}^{-1}$. Lastly, we have seen in the proof of Proposition E.3 that $R = 2h_{d-1}$, which finishes the proof. $\qquad\square$

## B.5 Proof of Theorem 4.7

*Proof.* We prove the statements in separate subsections. We consider interactions of maximum order $s_0 \geq 1$, summarized in $\mathcal{I} := \{ S \subset D \mid s = s_0 \}$. To show that SII and STI are s-efficient, by Theorem 4.1 it suffices to show that $\sum_{S \in \mathcal{I}} \gamma_s^m(t, |T \cap S|) = 0$ for all $T \in \mathcal{T}_{s_0}$. Given a subset $T \in \mathcal{T}_{s_0}$ with $|T \cap S| = k$ and $k \in \{0, \dots, s_0\}$, we have

$$\sum_{S \in \mathcal{I}} \gamma_s^m(t, |T \cap S|) = \sum_{k=0}^{s_0} \binom{t}{k} \binom{t-k}{s_0 - k} \gamma_s^m(t, k) = \sum_{k=0}^{s_0} (-1)^{s_0 - k} \binom{t}{k} \binom{d-t}{s_0 - k} m(t-k).$$

### B.5.1 Proof of s-efficiency for SII.

We let $m(t) := m_{s_0}^{\text{SII}}(t) = \frac{(d - t - s_0)! t!}{(d - s_0 + 1)!} = \frac{1}{d - s_0 + 1} \binom{d - s_0}{t}^{-1}$ and obtain

$$\sum_{S \in \mathcal{I}} \gamma_s^m(t, |T \cap S|) = \frac{1}{d - s_0 + 1} \sum_{k=0}^{s_0} (-1)^{s_0 - k} \binom{t}{k} \binom{d-t}{s_0 - k} \binom{d - s_0}{t - k}^{-1}.$$

We now consider

$$\sum_{k=0}^{s_0} (-1)^k \binom{t}{k} \binom{d-t}{s_0 - k} \binom{d - s_0}{t - k}^{-1} = \sum_{k=0}^{s_0} (-1)^k \frac{t!}{k!(t-k)!} \frac{(d-t)!}{(s_0 - k)!(d - t - s_0 + k)!} \frac{(t-k)!(d - s_0 - t + k)!}{(d - s_0)!}$$

$$= \frac{t!(d-t)!}{(d - s_0)!} \sum_{k=0}^{s_0} (-1)^k \frac{1}{(s_0 - k)! k!}$$

$$= \frac{t!(d-t)!}{s_0!(d - s_0)!} \sum_{k=0}^{s_0} (-1)^k \binom{s_0}{k}$$

$$= \frac{t!(d-t)!}{s_0!(d - s_0)!} (1 - 1)^{s_0}$$

$$= 0,$$

where we have used the binomial expansion for $(1 - 1)^{s_0}$. Hence,

$$\sum_{S \in \mathcal{I}} \gamma_s^m(t, |T \cap S|) = 0,$$

which finishes the proof of SII s-efficiency.

### B.5.2 Proof of s-efficiency of STI.

For STI, i.e. $m_{s_0}^{\text{STI}} := s_0 \frac{(d-t-1)!t!}{d!}$, we have

$$\sum_{S \in \mathcal{I}} \gamma_s^m(t, |T \cap S|) = \sum_{k=0}^{s_0} (-1)^{s_0-k} \binom{t}{k} \binom{d-t}{s_0-k} s_0 \frac{(d-t+k-1)!(t-k)!}{d!}$$

$$= \frac{s_0(-1)^{s_0} t!(d-t)!}{s_0! d!} \sum_{k=0}^{s_0} (-1)^k \binom{s_0}{k} \frac{(d-t+k-1)!}{(d-t-s_0+k)!}$$

As $\frac{(d-t+k-1)!}{(d-t-s_0+k)!}$ is a polynomial with orders less than $s_0$, we can use $\sum_{k=0}^{s_0} \binom{s_0}{k} k^m = 0$ for $m < s_0$ [38] to obtain

$$\frac{s_0(-1)^{s_0} t!(d-t)!}{s_0! d!} \sum_{k=0}^{s_0} (-1)^k \binom{s_0}{k} \frac{(d-t+k-1)!}{(d-t-s_0+k)!} = 0,$$

which finishes the proof of STI s-efficiency.

### B.5.3 Proof of efficiency for STI.

It is clear that s-efficiency of a CII implies by Theorem 4.1 with sampling order $k \geq s_0$ that the sum of top-order interaction estimates is

$$\sum_{S \in \mathcal{I}} \hat{I}_{s_0}^{\text{STI}}(t) = \sum_{S \in \mathcal{I}} c_{s_0}^{\text{STI}}(S).$$

On the other hand, it also implies that the sum of STI scores for the top-order interactions are

$$\sum_{S \in \mathcal{I}} I_{s_0}^{\text{STI}}(t) = \sum_{S \in \mathcal{I}} c_{s_0}^{\text{STI}}(S).$$

While lower-order estimates are computed exactly for STI, it follows that the sum of STI scores and sum of SHAP-IQ estimates over all interaction sets $\mathcal{S}_{s_0}$ are equal. Furthermore, due to the definition of STI, they must fulfill the efficiency axiom, which finishes the proof.

### B.5.4 Proof of efficiency for n-SII.

This result follows from the aggregation suggested by n-SII, which is independent of the index. The efficiency condition follows directly from the SV efficiency and the Bernoulli numbers, independent of higher-order interaction values, cf. proof in [3, Proposition 12]. The proof is based on induction, starting from the SV, where the efficiency condition holds. The specific aggregation based on the Bernoulli numbers then ensures that this efficiency condition is maintained. To apply this observation to SHAP-IQ estimates of n-SII, we observe that due to Proposition E.3, SHAP-IQ maintains the efficiency condition for order $s = 1$, i.e. the SV estimates. The aggregation of n-SII for the SHAP-IQ estimates of higher orders then immediately implies that this efficiency condition is always preserved, as the arguments presented therein hold independent of the interaction index, cf. [3, Proof of Proposition 12]. $\square$

# C   Algorithmic Details of SHAP-IQ

In this section, we decribe further algorithmic details that are used in the SHAP-IQ implementation.

## C.1   Algorithm of SHAP-IQ

The pseudo-code for SHAP-IQ is outlined in Algorithm 1.

---

**Algorithm 1** SHAP-IQ for any-order interactions $\mathcal{S}_{s_0}$ up to order $s_0$

---

**Require:** Budget $K > 0$, weights $q(t) \geq 0$ with $t \in [d]$, precomputed weights $\gamma_s^m(t, \ell)$ for $s = 1, \ldots, s_0, t = 0 \ldots, d$ and $\ell = 0, \ldots, s$

1: $k_0 \leftarrow \textsc{GetSamplingOrder}(q, K)$
2: **for** $T \notin \mathcal{T}_{k_0}$ **do**                                                            ▷ Deterministic
3:     $\eta \leftarrow \nu_0(T)$
4:     **for** $S \in \mathcal{S}_{s_0}$ **do**
5:         $c_{k_0}(S) \leftarrow c_{k_0}(S) + \eta \cdot \gamma_s^m(t, |T \cap S|)$          ▷ Update deterministic part for all $S$
6:     **end for**
7:     $K \leftarrow K - 1$
8: **end for**
9: **for** $t = k_0, \ldots, d - k_0$ **do**
10:     $p(t) \leftarrow q(t) / \left( \sum_{k=k_0}^{d-k_0} q(k) \binom{d}{k} \right)$          ▷ compute probabilities $\mathbb{P}_{k_0}(|T| = t)$
11: **end for**
12: **for** $k = 1, \ldots, K$ **do**                                                              ▷ Sampling
13:     $T \leftarrow \textsc{Sample}(p, k_0)$
14:     $\eta \leftarrow \nu_0(T)$
15:     **for** $S \in \mathcal{S}_{s_0}$ **do**
16:         $\Delta(S) \leftarrow \eta \cdot \gamma_s^m(t, |T \cap S|) \binom{d}{t} / p[t]$          ▷ Use probabilities
              $\mathbb{P}_{k_0}(T) = \mathbb{P}_{k_0}(|T| = t) / \binom{d}{t} \propto q(t)$
17:     **end for**
18:     $\hat{\mu}, \hat{s}^2 \leftarrow \textsc{WelfordUpdate}(\hat{\mu}, s2, k, \Delta)$
19: **end for**
20: mean $\hat{I}_{k_0}^m \leftarrow c_{k_0} + \hat{\mu}^m$ and variance $\hat{\sigma}^2 \leftarrow s2 / (n - 1)$
21: **return** $\hat{I}_{k_0}^m$ and $\hat{\sigma}^2$

---

## C.2   Sampling Weights $q(t)$

In our implementation, we rely on $q(t) \propto \mu(t)$ for $1 \leq t \leq d - 1$, where the weights $q(0) = q(d) = q_0 \gg 0$ are set to a high positive constant, which favors these subsets before weighting the remaining subsets in $\mathcal{T}_1$. These weights ensure that SHAP-IQ is equal to U-KSH for the SV. Another choice of weights is $q(t) = \frac{(d - t - s_0)!(t - s_0)!}{(d - s_0 + 1)!}$ for $s_0 \leq t \leq d - s_0$ and $q(t) = q_0$ otherwise, which prefers all orders up to $s_0$ and from $d$ to $d - s_0$. This choice of subset weights may be beneficial for very low budgets, as it is important to ensure that $k_0 \geq s_0$ for SHAP-IQ to maintain the efficiency condition. The algorithm to find $k_0$ given weight $q$ and budget $K$ is outlined in Algorithm 2. The sampling procedure to generate a subset according to $p(T)$ is outlined in Algorithm 3.

## C.3   Welford's Algorithm

Welford's algorithm [45] allows to iteratively update the mean and variance using a single pass. The algorithm is outlined in Algorithm 4.

**Algorithm 2** Determine the the sampling order $k_0$ for the deterministic part

**Require:** weights $q$ over $0, \ldots, d$, budget $K > 0$

1: initialize $k_0 = 0$
2: **for** $t = 0, \ldots, \text{FLOOR}(d/2)$ **do**
3:     $R = \left( \sum_{k=k_0}^{d-k_0} q[k]\binom{d}{k} \right)$                                           $\triangleright$ Normalization
4:     $p[t] \leftarrow q[t]\binom{d}{t}/R$
5:     $p[d-t] \leftarrow q[d-t]\binom{d}{t}/R$
6:     **if** $K \cdot q[t] > R$ and $K \cdot q[d-t] > R$ **then**
7:         $k_0 \leftarrow k_0 + 1$
8:         $K \leftarrow K - 2\binom{d}{t}$
9:     **end if**
10: **end for**
11: **return** $k_0$

---

**Algorithm 3** Sample a subset $T \sim p(T)$

**Require:** $p$ with $\sum_{k=k_0}^{d-k_0} p[k] = 1$, sampling order $k_0$

1: **for** $t = k_0, \ldots, d - k_0$ **do**
2:     $p(|T| = t) \leftarrow p[t]\binom{d}{t}$
3: **end for**
4: choose subset size $t_0$ with probability $p(|T| = t)$
5: choose subset $T$ of size $t_0$ with probability $\binom{d}{t_0}^{-1}$
6: **return** $T$

---

**Algorithm 4** Welford Algorithm for Mean and Variance [45]

**Require:** $\mu, s2, n, \Delta$
1: $n \leftarrow n + 1$
2: $\Delta_1 \leftarrow \Delta - \mu$
3: $\mu \leftarrow \mu + \Delta/n$
4: $\Delta_2 \leftarrow \Delta - \mu$
5: $s2 \leftarrow s2 + \Delta_1\Delta_2$
6: **return** $\mu, s2$

# D Experiments

For the interested reader, we provide a more detailed view on our empirical evaluation of Section 5. We give descriptions and pseudocode of the baseline algorithms approximating the three considered interaction indices SII, STI, and FSI, formal definitions of our synthetic games, and finally further obtained results that we omitted in the main part due to space constraints.

## D.1 Baseline Algorithms for SII, STI, and FSI

In this section, we describe our baseline algorithms for SII, STI and FSI. We distinguish between permutation-based approximation (SII and STI) and kernel-based approximation (FSI).

### D.1.1 Permutation-based (PB) Approximation

The algorithm for SII is outlined in Algorithm 5. Note that with each permutation only $d - s + 1$ interaction estimates of order $s$ are updated.

---

**Algorithm 5** Permutation-based sampling for SII for all orders up to $s_0$ [40]

---

**Require:** maximum interaction order $s_0$, interaction set $\mathcal{S}_{s_0}$, budget $K$
1: sum[S] $\leftarrow 0$ for all $S \in \mathcal{S}_{s_0}$
2: count[S] $\leftarrow 0$ for all $S \in \mathcal{S}_{s_0}$
3: permutationCost $\leftarrow 0$
4: **for** $s = 1, \ldots, s_0$ **do**
5:    permutationCost $\leftarrow$ permutationCost $+2^s \cdot (d - s + 1)$      ▷ Evaluate costs per permutation
6: **end for**
7: **while** $K \geq$ permutationCost **do**
8:    $\pi \leftarrow \{i_1, \ldots, i_d\}$ random permutation of $D$
9:    **for** $s = 1, \ldots, s_0$ **do**
10:      **for** $m = 1, \ldots, d - s + 1$ **do**
11:        $S \leftarrow \{i_m, \ldots, i_{m+s-1}\}$
12:        $T \leftarrow \{i_1, \ldots, i_{m-1}\}$ the set of predecessors of $i_m$ in $\pi$
13:        sum[S] $\leftarrow$ sum[S] $+ \delta_S^\nu(T)$        ▷ $\delta_S^\nu(T)$ costs $2^s$ evaluations
14:        count[S] = count[S]+1
15:      **end for**
16:    **end for**
17:    $K \leftarrow K-$ permutationCost      ▷ Update budget
18: **end while**
19: SII[S] $\leftarrow$ sum[S]/count[S] for all $S \in \mathcal{S}_{s_0}$.
20: **return** SII

---

The sampling-based algorithm for top-order interactions of STI is outlined in Algorithm 6. Note that with each permutation all top-order interaction estimates can be updated. However, the update requires a significant amount (permutationCost) of model evaluations.

### D.1.2 Kernel-based (KB) Approximation

Given a budget of $K$, we first find the sampling budget by identifying $k_0$ according to Algorithm 2 with weights $q(t) := \mu(t)$ and subtracting the number of subsets used for the deterministic part. We then sample the remaining subsets according to $p(T) \propto \mu(t)$ according to Algorithm 3.

**Algorithm 6** Permutation-based sampling for STI for all orders up to $s_0$ [39, 40]

---

**Require:** maximum interaction order $s_0$, interaction set $\mathcal{S}_{s_0}$, budget $K$

1: sum[S] $\leftarrow 0$ for all $S \in \mathcal{S}_{s_0}$
2: count[S] $\leftarrow 0$ for all $S \in \mathcal{S}_{s_0}$
3: **Compute exact (trivial) lower-order interactions**
4: eval[S] $\leftarrow 0$ for all $S \in \mathcal{S}_{s_0-1}$         ▷ Model evaluations for lower-order STI values
5: **for** $S \in \mathcal{S}_{s_0-1}$ **do**        ▷ Precompute model evaluations for lower-order STI
6:   eval[S] $\leftarrow \nu(S)$
7:   $K \leftarrow K - 1$
8: **end for**
9: **for** $S \in \mathcal{S}_{s_0-1}$ **do**              ▷ Lower-order interactions
10:   **for** $L \in \mathcal{P}(S)$ **do**
11:    SII[S] $\leftarrow$ SII[S] $+ (-1)^{s-l} \cdot$ eval[L]      ▷ Exact lower-order STI values
12:   **end for**
13: **end for**
14: **Compute sampling-based top-order interaction estimates**
15: permutationCost $\leftarrow 2^{s_0} \cdot \binom{d}{s_0}$     ▷ Every $S$ requires to compute $\delta_S^\nu$ with $2^{s_0}$ evaluations
16: **while** $K \geq$ permutationCost **do**         ▷ Evaluate one permutation
17:   $\pi \leftarrow \{i_1, \ldots, i_d\}$ random permutation of $D$
18:   **for** all top-order interactions $S$ **do**
19:    $i_m \leftarrow$ the leftmost element of $S$ in $\pi$
20:    $T \leftarrow \{i_1, \ldots, i_{m-1}\}$ the set of predecessors of $i_m$ in $\pi$
21:    sum[S] $\leftarrow$ sum[S] $+ \delta_S^\nu(T)$        ▷ $\delta_S^\nu(T)$ costs $2^{s_0}$ evaluations
22:    count[S] = count[S]+1
23:   **end for**
24:   $K \leftarrow K -$ permutationCost           ▷ Update budget
25: **end while**
26: STI[S] $\leftarrow$ sum[S]/count[S] for all top-order interactions.
27: **return** STI

---

Given the collection of $K$ subsets (deterministic and sampled), we solve an approximated weighted least square objective as

$$
\mathbb{E}_{T \sim p(T)} \left[ \left( \nu(T) - \sum_{\substack{S \in \mathcal{S}_{s_0} \\ S \subseteq T}} \beta(S) \right)^2 \right]
$$

$$
\approx \sum_{T \in \mathcal{T}_{k_0}} p(T) \left( \nu(T) - \sum_{\substack{S \in \mathcal{S}_{s_0} \\ S \subseteq T}} \beta(S) \right)^2 + p(T \in \mathcal{T}_{k_0}) \mathbb{E}_{T \sim p_{k_0}(T)} \left[ \left( \nu(T) - \sum_{\substack{S \in \mathcal{S}_{s_0} \\ S \subseteq T}} \beta(S) \right)^2 \right],
$$

where $k_0$ is found similar to SHAP-IQ and $p_{k_0}$ is a probability distribution over $\mathcal{T}_{k_0}$ with $p_{k_0}(T) \propto \mu(t)$, which is related to $p$ with $p_{k_0}(T) = p(T)/p(T \in \mathcal{T}_{k_0})$. The expectation over $p_{k_0}$ is then found by Monte Carlo integration. Approximating this objective yields a weighted sum of sampled subsets that approximates the weighted least square problem. This approximated least-square problem is then computed explicitly using

$$
\hat{I}^{\text{FSI}} = (\mathbf{Z}^T \mathbf{W} \mathbf{Z})^{-1} \mathbf{Z}^T \mathbf{W} \mathbf{y}, \tag{3}
$$

where $Z \in \{0,1\}^{K \times d_{s_0}}$ is a matrix that represents a binary encoding for each sampled subset where an entry in column $S \in \mathcal{S}_{s_0}$ is equal to one, if the subset contains $S$ and zero otherwise. The matrix $W \in \mathbb{R}^{K \times K}$ contains the weights for each subset on the diagonal, e.g. $p(T)$ for subsets of the deterministic part or $p(T \in \mathcal{T}_{k_0})/m$ for subsets of the sampled part, where $m$ refers to the number of sampled subsets for Monte Carlo integration. The vector $\mathbf{y}$ consists of all model evaluations $\nu_0(T)$, where $T$ is in the collection of subsets. To include the optimization constraint, we add $D$ to the

collection with weight set to a high positive constant (mimicking infinite). The algorithm is outlined in Algorithm 7.

---

**Algorithm 7** Kernel-based approximation of FSI [26, 40]

---

**Require:** maximum interaction order $s_0$, budget $K$, high constant $c_0 >> 0$.
 1: Weight vector $w[T]$ with one row and column per distinct subset $T$.
 2: Binary subset matrix $Z[T, S]$ with one row per distinct subset $T$ and one column per interaction subset $S$.
 3: Model evaluation vector $y[T]$ with model evaluations $\nu(T)$ per distinct subset $T$.
 4: Index array: $I$ per distinct subset $T$
 5: **Initialize constraints**
 6: $w \leftarrow \text{APPEND}(w, c_0)$
 7: $Z \leftarrow \text{APPENDROW}(Z, \mathbf{1}^T)$ with $\mathbf{1}^T$ of length $d_{s_0}$.
 8: $y \leftarrow \text{APPEND}(y, \nu_0(D))$.
 9: $K \leftarrow K - 1$.
10: **Deterministic Part**
11: **for** $t = 1, \dots, d - 1$ **do** $\qquad\qquad\qquad\qquad$ ▷ Initialize subset size probabilities as $\mathbb{P}_1(|T| = t)$
12: $\quad p(t) \leftarrow q(t) / \left( \sum_{k=1}^{d-1} q(k) \binom{d}{k} \right)$
13: **end for**
14: $k_0 \leftarrow \text{GETSAMPLINGORDER}(q, K)$
15: **for** $T \in \mathcal{T}_1$ and $T \notin \mathcal{T}_{k_0}$ **do** $\qquad\qquad\qquad\qquad\qquad\qquad\qquad$ ▷ Deterministic
16: $\quad Z \leftarrow \text{APPENDROW}(Z, \text{BINARY}(T))$
17: $\quad y \leftarrow \text{APPEND}(y, \nu_0(T))$
18: $\quad w \leftarrow \text{APPEND}(w, p(t)/\binom{d}{t}) \qquad\qquad$ ▷ Weight with probability $\mathbb{P}_1(T) = \mathbb{P}_1(|T| = t)/\binom{d}{t}$
19: $\quad K \leftarrow K - 1$
20: **end for**
21: **Sampling Part**
22: $w_0 \leftarrow sum(p(t))$ for $k_0 \leq t \leq d - k_0$. $\qquad\qquad\qquad\qquad$ ▷ Remaining probability weight
23: **for** $k = 1, \dots, K$ **do** $\qquad\qquad\qquad\qquad\qquad\qquad\qquad\qquad\qquad\qquad$ ▷ Sampling
24: $\quad T \leftarrow \text{SAMPLE}(p, k_0)$
25: $\quad$ **if** $T \in I$ **then**
26: $\quad\quad Z \leftarrow \text{APPENDROW}(Z, \text{BINARY}(\text{T}))$.
27: $\quad\quad y \leftarrow \text{APPEND}(y, \nu_0(T))$
28: $\quad\quad w \leftarrow \text{APPEND}(w, 1)$
29: $\quad\quad I.\text{ADDINDEX}(T)$
30: $\quad$ **else**
31: $\quad\quad w[I[T]] \leftarrow w[I[T]] + 1$
32: $\quad$ **end if**
33: **end for**
34: $w[I[T]] \leftarrow w[I[T]] \cdot w_0/K$ for all $T \in I$ $\qquad\qquad\qquad\qquad\qquad\qquad$ ▷ Rescaling
35: $W \leftarrow \text{DIAG}(w)$ $\qquad\qquad\qquad\qquad\qquad\qquad\qquad$ ▷ Diagonal matrix with diagonal $w$
36: $\text{FSI} \leftarrow \text{SOLVEWLS}(Z, W, y)$.
37: **return** FSI

---

### D.1.3 Computational Complexity of Baseline Methods

To evaluate one permutation for STI, the PB algorithm requires $2^s$ model evaluations per interaction, i.e. in total $\binom{d}{s_0} \cdot 2^{s_0}$ for all top-order interactions. With each evaluated permutation all interaction estimates can be updated. For lower order interactions, STI requires to compute model evaluations for all subsets with $|S| \leq s_0$. For SII, the complexity is $(d - s + 1) \cdot 2^s$ per permutation and only interaction estimates with $S \in \pi$ can be updated per permutation, i.e. $d - s + 1$ interaction estimates with one permutation. This constitutes a significant drawback over the PB approximations for SV, which iterates only once through the permutation requiring $d - 1$ evaluations to update all estimates of the SV. In contrast, as for SV, the KB approach of FSI allows to update all interaction estimates using *one single* model evaluation. However, the KB approach of FSI always requires to estimate *all* interactions with order $s \leq s_0$ and its computational effort increases non-linear with the number of subsets used, as solving the weighted least square problem requires inverting a $K \times d_{s_0}$ matrix.

## D.2 Further Information about the Models

This section contains a detailed information about the models (SOUM, LM, and ICM) used in our experiments.

### D.2.1 Sum of Unanimity Model (SOUM)

**Definition D.1** (Sum of Unanimity Model (SOUM)). *For $N$ subsets $Q_1, \ldots, Q_N \subseteq D$ and coefficients $a_1, \ldots, a_N \in \mathbb{R}$ the* sum of unanimity model (SOUM) *is defined as*

$$\nu(T) := \sum_{n=1}^{N} a_n \mathbf{1}(Q_n \subseteq T).$$

For SOUMs, it is possible to efficiently compute the ground-truth values for CII.

**Proposition D.2** (Ground-truth values for SOUM). *For a SOUM, it holds*

$$I_\nu^m(S) = \sum_{n=1}^{N} a_n \omega(q_n, |S \cap Q_n|),$$

*with*

$$\omega(q, r) = \sum_{t=q}^{d} \sum_{k=0}^{k_{max}(r)} \binom{d - q - (s - r)}{t - q - k} \binom{s - r}{k} \gamma_s^m(t, k + r)$$

*and $k_{max}(r) := \min(t - q, s - r)$*

*Proof.* Due to the linearity of the CII, it suffices to compute the CII for $\nu_Q(T) := \mathbf{1}(Q \subseteq T)$. By Theorem 4.1, we have

$$
\begin{aligned}
I_{\nu_Q}^m(S) &= \sum_{T \subseteq D} \mathbf{1}(Q \subseteq T) \gamma_s^m(t, |T \cap S|) \\
&= \sum_{t=q}^{d} \sum_{k=0}^{k_{max}} \binom{d - q - (s - |S \cap Q|)}{t - q - k} \binom{s - |S \cap Q|}{k} \gamma_s^m(t, k + |S \cap Q|) \\
&=: \omega(q, |S \cap Q|),
\end{aligned}
$$

where we used that $Q \cap S \subseteq T \cap S$ due to $\mathbf{1}(Q \subseteq T)$ and $|T \cap S| = |S \cap Q| + (|T \cap S|) \setminus (|S \cap Q|)$, where $k := |(T \cap S) \setminus (S \cap Q)|$ ranges from 0 to $k_{max} := \min(t - q, s - |S \cap Q|)$. Since $S \cap Q$ is fixed, we need to count the number of subsets $T$ of size $t$, given $k$, such that $|T \cap S| = |S \cap Q| + k$. We count $\binom{s - |S \cap Q|}{k}$ ways to choose subsets of elements that are not in $S \cap Q$ but are in $S$. Then $q - (s - |S \cap Q|)$ elements of $T$ are fixed. We thus select from $d - q - (s - |S \cap Q|)$ elements exactly $t - q - k$ elements, as $q$ and $k$ elements are already contained in $T$.

Finally, the CII value is given as

$$I_\nu^m(S) = \sum_{n=1}^{N} a_n \omega(q_n, |S \cap Q_n|),$$

where the weights $\omega$ can be precomputed with $|S \cap Q_n| \in \{0, \ldots, s\}$. $\qquad\square$

### D.2.2 Language Model (LM)

For a language model (LM), we use a fine-tuned version of the DistilBERT[5] transformer architecture [34] on movie review sentences from the original *IMDB* dataset [27] for sentiment analysis, i.e. $\nu$ has values in $[-1, 1]$. The IMDB data stems from the *dataset* library [22]. In the LM, for a given sentence, different feature coalitions are computed by masking absent features in a tokenized sentence. The implementation is based on the *transformers* API [47].

---

[5]The model can be found at `https://huggingface.co/dhlee347/distilbert-imdb`.

### D.2.3 Image Classifier (ICM)

The image classification model (ICM) is a ResNet18 [19] pre-trained on ImageNet [11] as provided by *torch* [30]. We randomly sample $50$ images from ImageNet [11] and explain the prediction of the highest probability class given the original image. To obtain the prediction of different coalitions, we pre-compute super-pixels with SLIC [1, 44] to obtain a function on $d = 14$ features and apply mean imputation on absent features.

### D.3 Hardware Details and Environmental Impact of the Experiments

Running the experiments required approximately $2\,000$ CPU hours in total. The experiments concerning the approximation quality of SHAP-IQ compared to the baselines were run on an computation cluster on hyperthreaded Intel Xeon E5-2697 v3 CPUs clocking at with 2.6Ghz. To further increase the efficiency of the experiments, the outputs of the LM and ICM were pre-computed given the powerset of all features. Around $1\,500$ CPU hours were consumed for these experiments on the cluster. Before running the experiments on the cluster, the implementations were validated on a Dell XPS 15 9510 containing an Intel i7-11800H at 2.30GHz. For this and further small-scale experiments like the n-SII values approximately $500$ CPU hours were consumed.

### D.4 Further Experimental Results

This section describes the further results and experiments omitted in the main body of the work.

### D.4.1 Approximation Quality of top Order Interactions.

We further compute interaction scores for $s_0 = 1$, $s_0 = 2$, $s_0 = 3$, and $s_0 = 4$ of all three interaction indices SII, STI, and FSI on the LM. We plot the MSE and Prec@10 based on $g = 50$ independent iterations for these settings. All results are summarized in Figure 6. Moreover, we compute interaction scores for $s_0 = 1$, $s_0 = 2$, $s_0 = 3$, and $s_0 = 4$ of all three interaction indices SII, STI, and FSI on the ICM. The MSE and Prec@10 based on $g = 50$ independent iterations for these settings are shown in Figure 7. Further results to the plots show in Section 5.2 for the SOUM are presented in Figure 8.

### D.4.2 n-SII Estimation on Example Sentences.

We further probe the LM with randomly selected sentences from the *IMDB* dataset and estimate the SII scores up to order $4$. For the example sentence "*It is a gruesome cannibal movie. But it's not bad. If you like Hannibal, you'll love this.*" in Section 5.2 all orders of n-SII with $s_0 = 1, 2, 3, 4$ are illustrated in Figure 9. Another example sentence "*I have never forgot this movie. All these years and it has remained in my life.*" with all orders of n-SII up to the maximum interaction order $s_0 = 1, 2, 3, 4$ is shown in Figure 10. Lastly, for four sentences n-SII for all orders with $s_0 = 4$ are visualized in Figure 11.

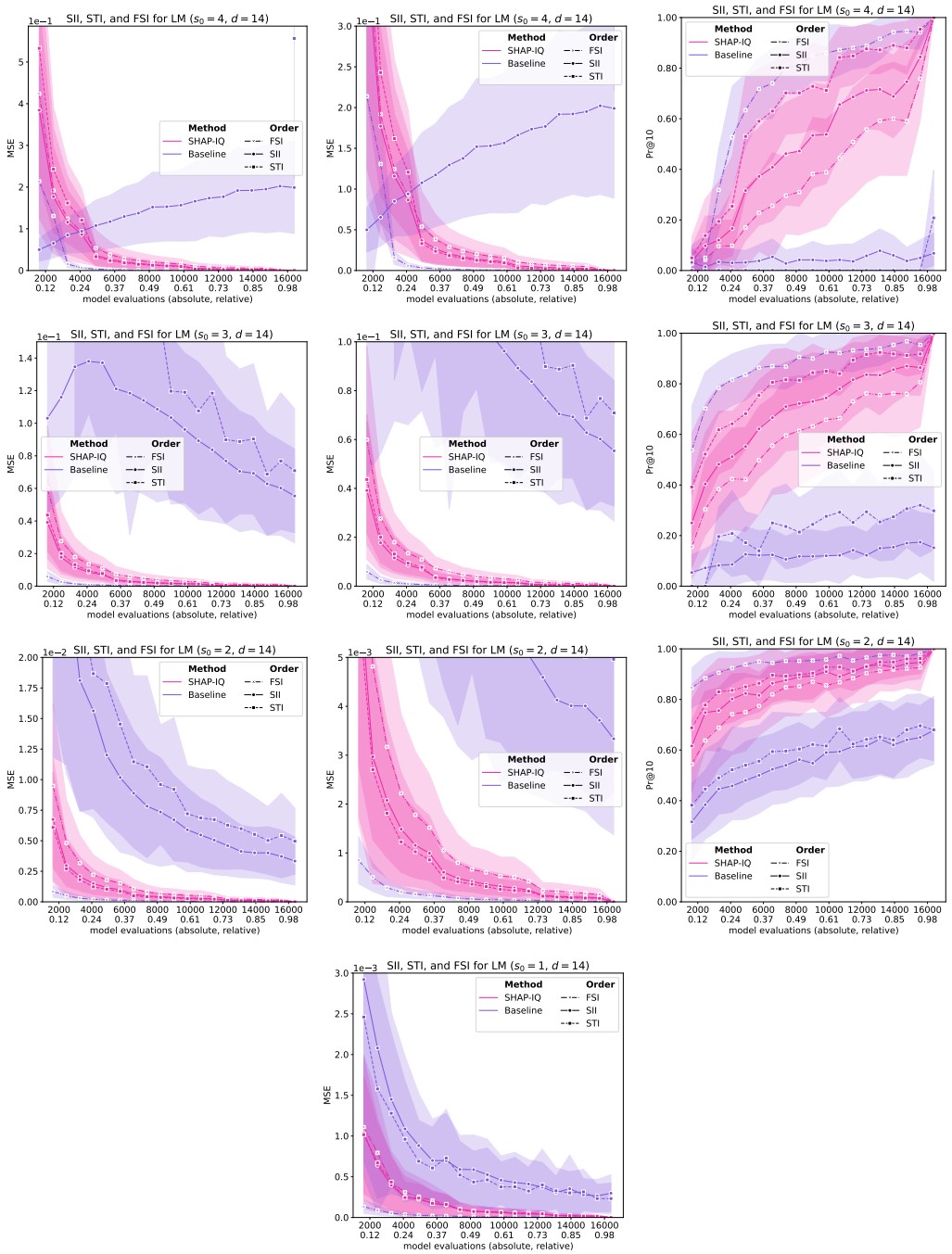

Figure 6: Approximation Quality for LM with interaction order $s_0 = 4$ for $g = 50$ iterations (first row), with interaction order $s_0 = 3$ for $g = 50$ iterations (second row), with interaction order $s_0 = 2$ for $g = 50$ iterations (third row), and with interaction order $s_0 = 1$ (Shapley Value) for $g = 50$ iterations (fourth row).

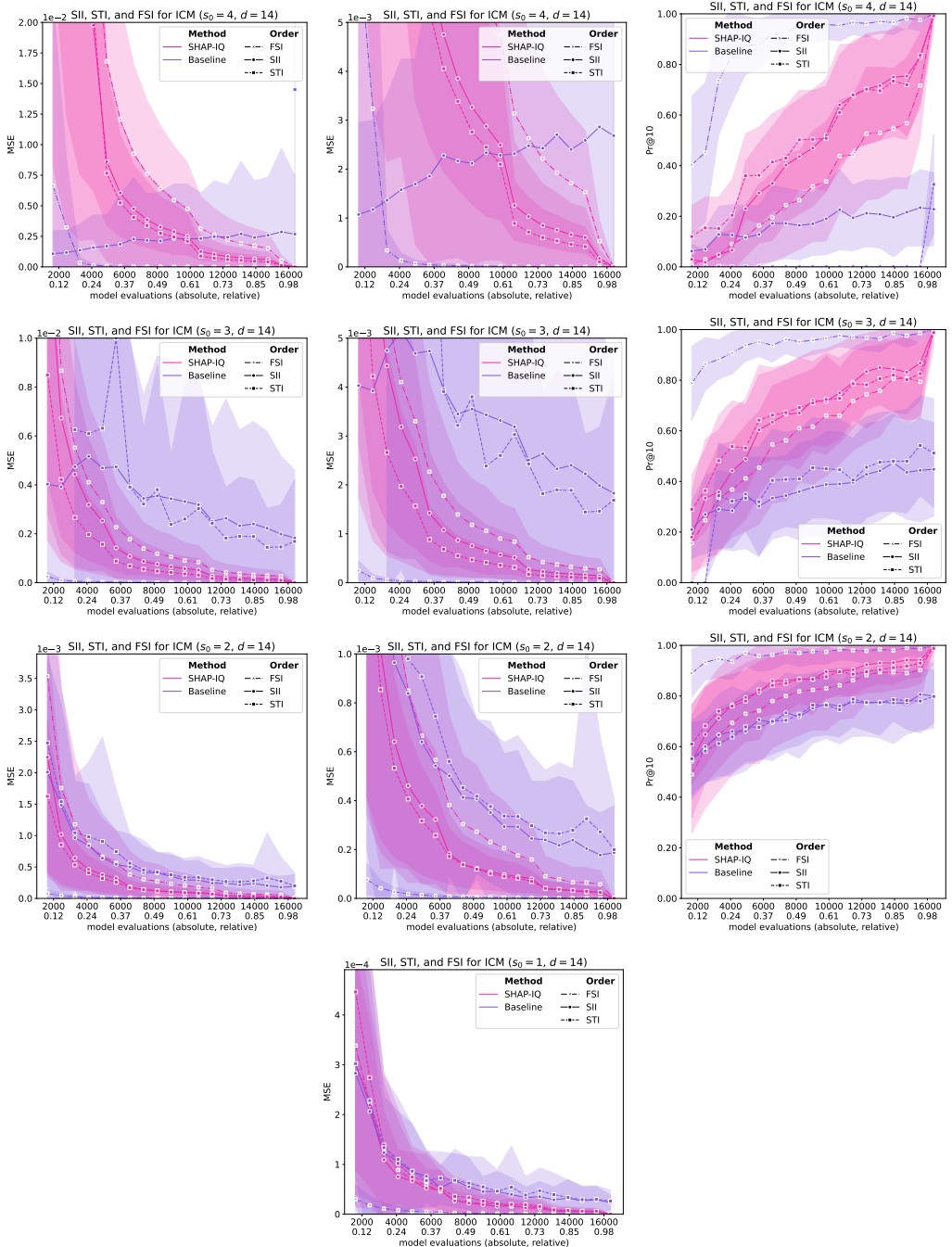

Figure 7: Approximation Quality for ICM with interaction order $s_0 = 4$ for $g = 50$ iterations (first row), with interaction order $s_0 = 3$ for $g = 50$ iterations (second row), with interaction order $s_0 = 2$ for $g = 50$ iterations (third row), and with interaction order $s_0 = 1$ (Shapley Value) for $g = 50$ iterations (fourth row).

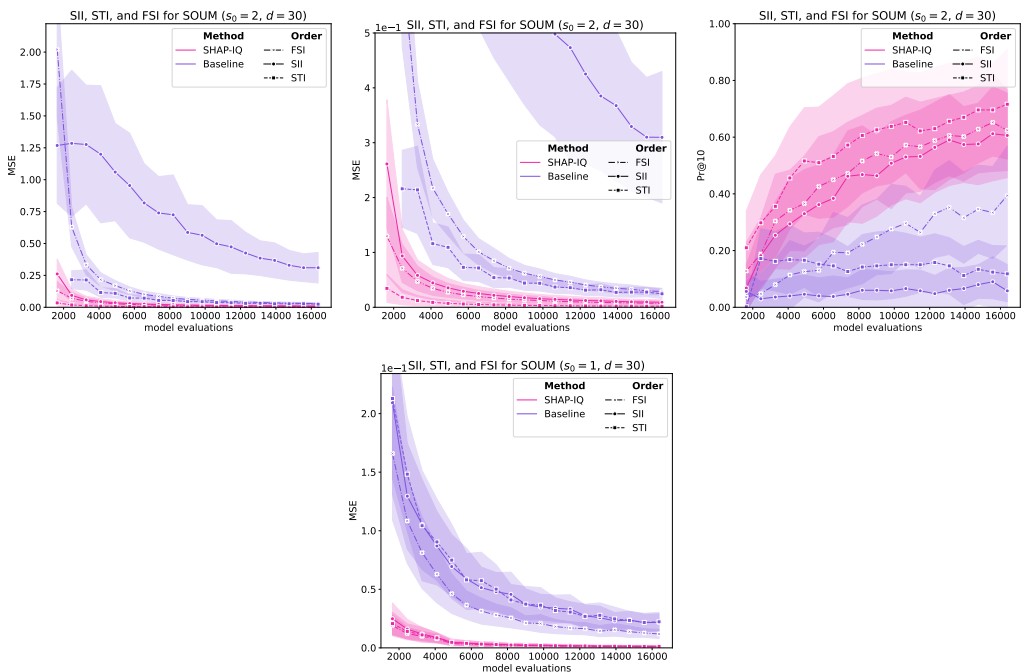

Figure 8: Approximation Quality for SOUM order $s_0 = 2$ (first row) and $s_0 = 1$ (Shapley value, second row) for $g = 50$ iterations on the SOUM with $N = 100$ interactions, $d = 30$ features, and $\ell_{\max} = 30$.

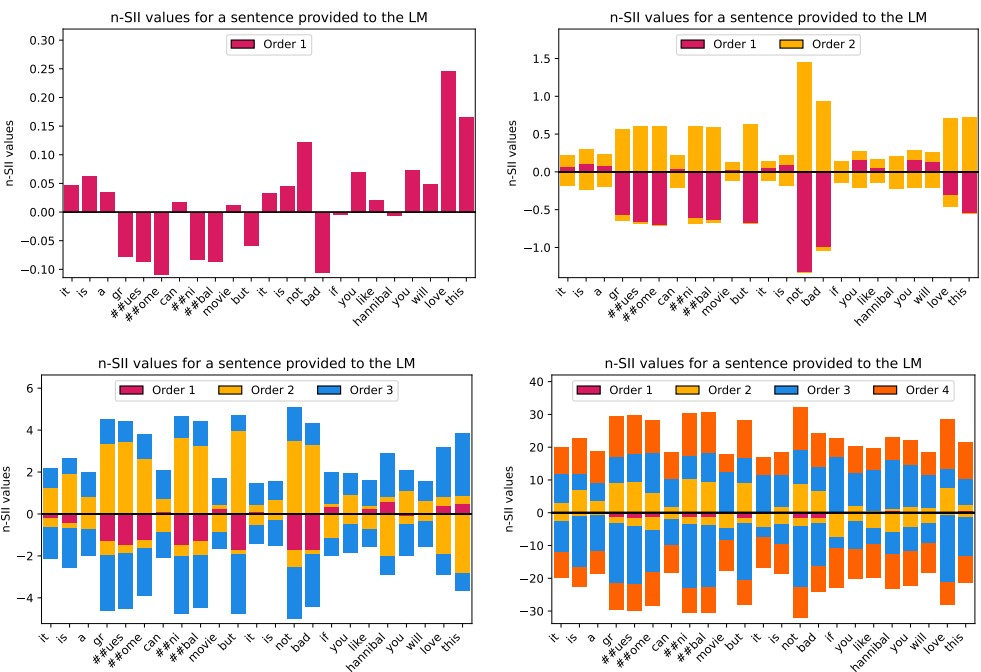

Figure 9: Estimated SII values with orders $s = 1, 2, 3, 4$ for the sentence "*It is a gruesome cannibal movie. But it's not bad. If you like Hannibal, you'll love this.*" ($d = 23$) provided to the LM. The plots show all orders of n-SII with maximum interaction order $s_0 = 1$ (top left), $s_0 = 2$ (top right), $s_0 = 3$ (bottom left), and $s_0 = 4$ (bottom right).

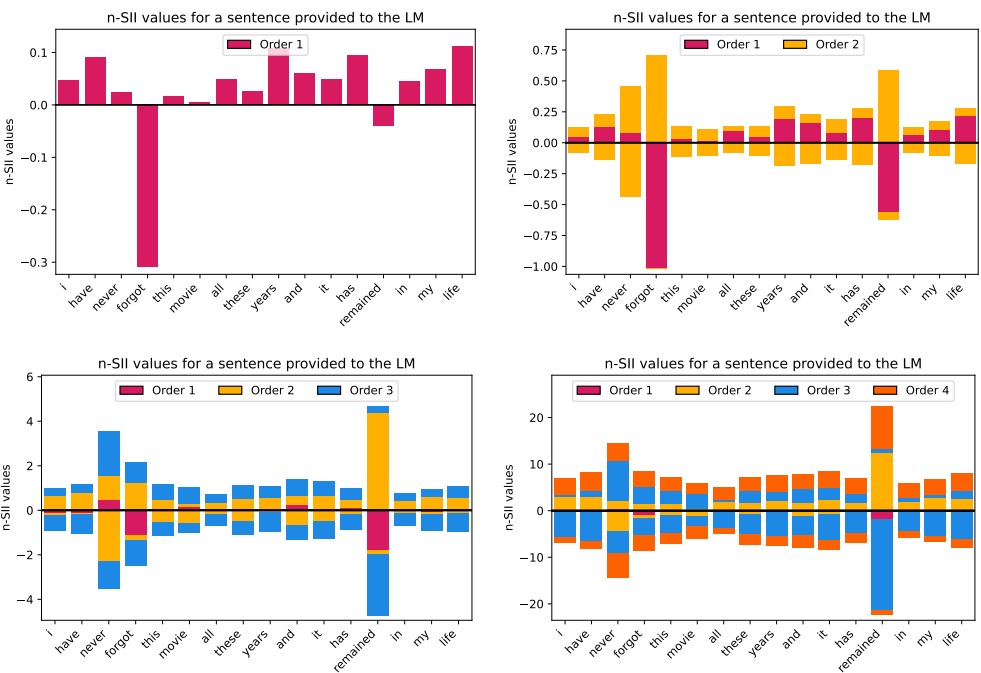

Figure 10: Estimated SII values with orders $s = 1, 2, 3, 4$ for the sentence "*I have never forgot this movie. All these years and it has remained in my life.*" ($d = 16$) provided to the LM. The plots show all orders of n-SII with maximum interaction order $s_0 = 1$ (top left), $s_0 = 2$ (top right), $s_0 = 3$ (bottom left), and $s_0 = 4$ (bottom right).

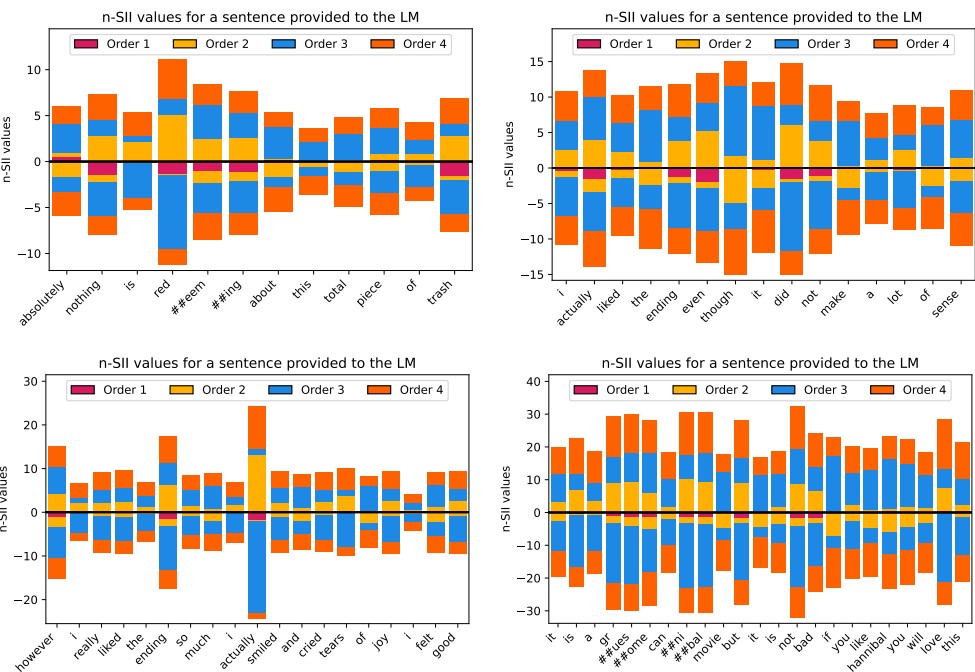

Figure 11: Estimated n-SII values with $s_0 = 4$ for the sentences provided to the LM: "*Absolutely nothing is redeeming about this total piece of trash.*" ($d = 12$, top left), "*I actually liked the ending even though it did not make a lot of sense.*" ($d = 15$, top right), "*However, I really liked the ending so much, I actually smiled and cried tears of joy. I felt good.*" with ($d = 19$, bottom left), and "*It is a gruesome cannibal movie. But it's not bad. If you like Hannibal, you'll love this.*" ($d = 23$, bottom right).

# E Further theoretical Results for the Shapley Value

In this section, we give two additional results for the special case of the SV. We explicitly state the inverse of the covariance matrix from [8] and further present a simplified representation of SHAP-IQ, if applied on the SV, which aligns with Theorem 4.4.

## E.1 Explicit Inverse of Covariance Matrix of Unbiased KernelSHAP

The covariance matrix has been explicitly computed in [8, Appendix A]. In this section, we provide the explicit form of the inverse $A^{-1}$. This inverse admits the same structure and is a central element in the explicit calculation of U-KSH as a weighted sum, which is linked to SHAP-IQ.

**Proposition E.1** (Explicit covariance matrix [8]). *For the covariance matrix it holds*

$$A := \mathbb{E}[ZZ^T] = \mu_2 \mathbf{J} + (\mu_1 - \mu_2)\mathbf{I},$$

*with $i, j \in D$ and constants*

$$A_{ii} := \mu_1 := \mathbb{P}(Z_i = 1) = \frac{1}{2} \qquad A_{ij} := \mu_2 := \mathbb{P}(Z_i = Z_j = 1) = \frac{1}{d(d-1)} \frac{\sum_{k=2}^{d-1} \frac{k-1}{d-k}}{\sum_{k=1}^{d-1} \frac{1}{k(d-k)}}.$$

*Proof.* The proof is given in [8, Appendix A]. $\qquad\qquad\square$

We have for $\sum_{k=1}^{d-1} \frac{1}{k(d-k)} = 2h_{d-1}$, where $h_n$ is the n-th harmonic number. Furthermore $\mu_1 - \mu_2 = \frac{1}{2h_{d-1}}$. We now give the explicit form of $A^{-1}$.

**Proposition E.2** (Explicit inverse of covariance matrix). *Let $A := \mathbb{E}[ZZ^T]$. Then, we have an explicit form of the inverse $A^{-1}$ as*

$$A^{-1} = \tilde{\mu}_2 \mathbf{J} + (\tilde{\mu}_1 - \tilde{\mu}_2)\mathbf{I}$$

*with constants*

$$(A^{-1})_{ii} := \tilde{\mu}_1 = 2h_{d-1}\frac{\mu_1 + (d-2)\mu_2}{\mu_1 + (d-1)\mu_2} \qquad (A^{-1})_{ij} := \tilde{\mu}_2 = 2h_{d-1}\frac{-\mu_2}{\mu_1 + (d-1)\mu_2}$$

*Proof.* The proof follows directly from Proposition E.1 and Lemma B.2. $\qquad\square$

## E.2 Simplified Representation of SHAP-IQ for the Shapley Value

In this section, give an explicit form of SHAP-IQ for the SV that admits a similar form as the SV representation in Theorem 4.4. We consider the SV weights $m(t) := \frac{(d-t-1)!t!}{d!}$.

**Proposition E.3** (SHAP-IQ for SV). *For SHAP-IQ with $p(T) \propto \mu(t)$ and sampling order $k_0 = 1$, it holds*

$$\hat{I}_1^m(i) = c_1(i) + \frac{2h_{d-1}}{K} \sum_{k=1}^{K} \nu_0(T_k)\left[\mathbf{1}(i \in T_k) - \frac{t_k}{d}\right],$$

*where $h_n$ is the $n$-th harmonic number.*

*Proof.* Recall the definition of SHAP-IQ of order 1

$$\hat{I}_1^m(i) := c_1(i) + \frac{1}{K} \cdot \sum_{k=1}^{K} \nu(T_k)\frac{\gamma_s^m(t_k, |T_k \cap \{i\}|)}{p(T_k)}.$$

with $p(T_k) := \mu(t_k)/R \propto \mu(t_k)$. We proceed by rewriting $\gamma_s^m(t_k, |T_k \cap \{i\}|) = \gamma_s^m(t_k, \mathbf{1}(i \in T_k))$ for $T_k \in \mathcal{T}_1$ as

$$\gamma_s^m(t_k, \mathbf{1}(i \in T_k)) = (-1)^{1-\mathbf{1}(i \in T_k)}\frac{(d-t-1+\mathbf{1}(i \in T_k))!(t-\mathbf{1}(i \in T_k))!}{d!}$$

$$= \mu(t)\frac{1}{d}\left[\mathbf{1}(i \in T_k)(d-t_k) - \mathbf{1}(i \notin T_k)t_k\right]$$

$$= \mu(t)\left[\mathbf{1}(i \in T_k) - \frac{t_k}{d}\right].$$

Hence,

$$\hat{I}_1^m(i) := c_1(i) + \frac{1}{K} \cdot \sum_{k=1}^{K} \nu(T_k) \frac{\mu(t) \left[ \mathbf{1}(i \in T_k) - \frac{t_k}{d} \right]}{p(T_k)}$$

$$= c_1(i) + \frac{R}{K} \cdot \sum_{k=1}^{K} \nu(T_k) \left[ \mathbf{1}(i \in T_k) - \frac{t_k}{d} \right].$$

For the normalizing constant, we have

$$R = \sum_{T \in \mathcal{T}_1} \mu(t) = \sum_{t=1}^{d-1} \mu(t) \binom{d}{t} = \sum_{t=1}^{d-1} \frac{d}{t(d-t)} = \sum_{t=1}^{d-1} \left( \frac{1}{t} + \frac{1}{d-t} \right) = 2h_{d-1},$$

which finishes the proof. $\qquad\square$

# F  Approximation Methods for the SV

There are two prominent representations of the SV, which are used for sampling-based approximation. Both allow to update all SVs simultaneously with one sample as well as maintaining the efficiency property.

## F.1  Permutation-based (PB) Approximation

Permutation-based (PB) approximation was introduced for SV [5]. It is based on the observation that the marginal contributions $\delta_i^\nu(T) = \nu(T \cup \{i\}) - \nu(T)$ can be computed by using permutations $\pi \in \mathfrak{S}_D$ of $D$ and $\nu(u_i^-(\pi)) - \nu(u_i^+(\pi))$, where $u_i^-(\pi), u_i^+(\pi)$ are the sets that consist of all elements preceding $i$ in $\pi$ with and without $i$, respectively. For for each subset $T \subseteq D \setminus \{i\}$ of size $t$ there are exactly $t!(d - t - 1)!$ permutations with $T = u_i^-(\pi)$ and thus

$$I^{\mathrm{SV}}(i) = \frac{1}{d!} \sum_{\pi \in \mathfrak{S}_D} \delta_i^\nu(u_S^-(\pi)) = \mathbb{E}_{\pi \sim \mathrm{unif}(\mathfrak{S}_D)}[\delta_i^\nu(u_S^-(\pi))].$$

This expectation can be efficiently approximated by sampling $\pi \sim \mathrm{unif}(\mathfrak{S}_D)$ and using a Monte Carlo estimate for the expectation. As $\sum_{i \in D} \delta_i^\nu(u_i^-(\pi)) = \nu(D) - \nu(\emptyset)$ for arbitrary permutations $\pi$, the efficiency constraint is maintained throughout the sampling procedure. The Monte Carlo estimates allows to apply well-established statistical results to obtain bounds on the approximation error [4].

## F.2  Kernel-based (KB) Approximation

Kernel Shapley Additive Explanation Values [26], short KernelSHAP (KSH), and Unbiased KernelSHAP (U-KSH) [8] make use of the representation of the SV as the solution to a constrained quadratic optimization problem [6]

$$I^{\mathrm{SV}} = \arg\min_\beta \sum_{T \in \mathcal{T}_1} \mu(t) \left( \nu_0(T) - \sum_{i \in T} \beta_i \right)^2 \tag{4}$$
$$\text{s.t.} \sum_{i \in D} \beta_i = \nu_0(D)$$

with $\nu_0(T) := \nu(T) - \nu(\emptyset)$, $\mathcal{T}_k := \{T \subseteq D : k \le t \le d - k\}$ and $\mu(t) := \frac{1}{d-1} \binom{d-2}{t-1}^{-1}$. This quadratic optimization problem can be solved explicitly using the weighted least square solution

$$I^{\mathrm{SV}} = (\mathbf{Z}^T \mathbf{W} \mathbf{Z})^{-1} \mathbf{Z}^T \mathbf{W} \mathbf{y}, \tag{5}$$

where $\mathbf{Z} \in \{0, 1\}^{2^d \times d}$ is a row-wise binary encoding of all subsets of $T \subseteq D$, $\mathbf{W} \in \mathbb{R}^{2^d \times 2^d}$ is a diagonal matrix with the subset weights $\mu$ and $\mathbf{y}$ consists of the evaluations of $\nu_0(T)$ for each subset. To include the optimization constraint, the (otherwise undefined) weights $\mu(d), \mu(0)$ of $D$ and $\emptyset$ are set to a high positive constant. Solving (5) still requires $2^d$ model evaluations and thus KSH [26] approximates $I^{\mathrm{SV}}$ by considering (5) as an expectation

$$\sum_{T \in \mathcal{T}_1} \mu(t) \left( \nu_0(T) - \sum_{i \in T} \beta_i \right)^2 \propto \mathbb{E}_{T \sim p(T)} \left[ \left( \nu_0(T) - \sum_{i \in T} \beta_i \right)^2 \right]$$

with $p(T) \propto \mu(t)$. Note that the optimization problem is invariant in terms of scaling. This expectation is the approximated similarly to SHAP-IQ, by computing high and low subset sizes explicitly and using Monte Carlo integration for the center sizes. The KSH estimator is difficult to analyze and it is only known that it is asymptotically unbiased [46].

KSH constructs a collection of subsets by determining a *sampling order* $k_0$, such that subsets with $k_0 \le t \le d - k_0$ are sampled from $p(T) \propto \mu(t)$ and for $t < k_0$ or $t > d - k_0$ all possible subsets are used. The value of $k_0$ is thereby found by successively comparing the expected number of subsets with the total number of subsets of that size. As the number of subsets $\binom{d}{t}$ for fixed $d$ is a symmetric

log-concave sequence of positive terms, it has a maximum at the middle term(s) $\lfloor \frac{d}{2} \rfloor, \lceil \frac{d}{2} \rceil$ and grows monotonically and symmetrically as $\binom{d}{t} = \binom{d}{d-t}$ towards this maximum from both sides. Thus, the implementation starts the comparison at $k_0 = 0$ and iteratively increases the $k_0$ candidate.

