# OpenReview forum: "SHAP-IQ: Unified Approximation of any-order Shapley Interactions"
_NeurIPS.cc/2023/Conference — NeurIPS 2023 poster_

### Official Review · Reviewer_VdQo · 2023-06-25

**Soundness:** 4 excellent
**Presentation:** 4 excellent
**Contribution:** 3 good
**Rating:** 7
**Confidence:** 3

**Summary:**

This paper provided an unified perspective to several existing Shapley-based interaction attribution methods as Cardinal Interaction Index (CII), in addition provide a more efficient sampling based approximation method. Connection between their formulation with unbiased KernelSHAP is established and they showed that their formulation can also improve the standard SHAP algorithm.

**Strengths:**

The unified formulation of several interaction attribution methods as CII is novel to the best of my knowledge in the X-AI literature. The authors made their claims and novelty very clear as well.

**Weaknesses:**

I think the content of the paper is very well written so not much weakness to comment. One thing that tricked me a bit is in line 33-34 when first mentioned CII it would be nice to have a citation to show that this is not your original contribution. Also in line 129 the \mathcal{S}_{s_0} notation is not defined properly so that stopped my flow a bit too.

**Questions:**

Not really from my side.

**Limitations:**

Not really from my side. Potential improvements are stated in the weakness section.

---

> ### Author Rebuttal · Authors · 2023-08-08
>
> We sincerely thank the anonymous reviewer for carefully reading and appreciation of our submitted work.
>
> We will follow your suggestions and add the reference for CII [11] already in line 33-34 and refer to the definition of $\mathcal S_{s_0}$ (given in line 84) in line 129 to improve readability.

---

### Official Review · Reviewer_cGA2 · 2023-07-03

**Soundness:** 3 good
**Presentation:** 4 excellent
**Contribution:** 3 good
**Rating:** 8
**Confidence:** 4

**Summary:**

This paper presents a unified approximation technique for Shapley interaction indices. It reformulates the general form of various Shapley interaction methods (Shapley interaction index, Shapley-Taylor, Faith-Shap) from a sum involving the set in question to a sum not involving the set, but instead a computation involving the method weights. This allows one call of the function to be a Monte Carlo integration sample of every set of inputs the method is calculating a value for. Theoretical guarantees of unbiasedness and efficiency are provided, as well as probabilistic error bounds.

**Strengths:**

Clear writing, very easy to follow and presented plainly.
Interesting and practical results which apply to multiple methods.
improvements over baseline is apparent in experiments.
The results are general enough that they are likely to apply to future methods.


**Weaknesses:**

Faith-Shap not fully addressed. But this seems to be in part to the inherent differences of that method.

**Questions:**

-Could you clarify, is this method an application of a technique used to approximate Shaley Values, but applied to interaction indices? If so, would you point out where this is is stated. If not, could you elaborate on the novel contributions?

Minor issues:
(153) Please define $\mathcal{G}_D$.
(130-132)_ Please clarify, this secion is dificult to follow.
(174) - the commas and grammar are off in this sentence.

**Limitations:**

Application to Faith-Shap as mentioned above.
The method applies to to a type of interaction form that is fairly general.

---

> ### Author Rebuttal · Authors · 2023-08-08
>
> We sincerely thank the anonymous reviewer for carefully reading and appreciation of our submitted work as highly relevant in terms of theoretical and empirical contributions.
>
> - Could you clarify, is this method an application of a technique used to approximate Shaley Values, but applied to interaction indices? If so, would you point out where this is is stated. If not, could you elaborate on the novel contributions?
>
> SHAP-IQ is directly motivated as a Monte Carlo approximation of the novel representation of the CII in Theorem 4.1.
> We found that in the special case of Shapley values, it directly corresponds to Unbiased KernelSHAP (U-KSH), cf. Theorem 4.5, and can, thus, be seen as a generalization of U-KSH with a greatly simplified computation (cf. line 330-332).
> However, our motivation based on Monte Carlo sampling differs fundamentally from the motivation of U-KSH, which relies on the weighted least squares representation, similar to KernelSHAP.
> We would further like to point out that in case of CIIs the motivation of U-KSH, similar to KernelSHAP, is impossible to utilize, as the representation as a weighted least square problem is unclear (cf. line 321-323).
>
> - Minor issues: (153) Please define $\mathfrak{G}_D$. (130-132) Please clarify, this secion is dificult to follow. (174) - the commas and grammar are off in this sentence.
>
> Thank you for pointing out the minor improvements in our manuscript, which we will address.

---

> > ### Comment · Reviewer_cGA2 · 2023-08-16
> > **Response to Author**
> >
> > We have read the rebuttal and thank the author for the response. We will not change our score.

---

### Official Review · Reviewer_xK2B · 2023-07-04

**Soundness:** 3 good
**Presentation:** 3 good
**Contribution:** 3 good
**Rating:** 7
**Confidence:** 4

**Summary:**

This paper presents a novel method, SHAPley Interaction Quantification (SHAP-IQ), for efficiently computing Shapley interaction indices in the context of explainable artificial intelligence (XAI). Shapley interaction indices extend the concept of Shapley values, a method from cooperative game theory, to measure the contribution of interactions between features in a model. The authors propose SHAP-IQ as a unified sampling-based approximation method that can be applied to any Cardinal Interaction Index (CII). They provide theoretical guarantees for SHAP-IQ, showing that it is unbiased and consistent, and provide a general approximation bound. They also discuss the computational complexity of SHAP-IQ and how it compares to other methods. The paper also presents a new representation for the Shapley value (SV) and shows that in the special case of single feature subsets, SHAP-IQ provides novel insights into the SV and greatly simplifies the calculation of Unbiased KernelSHAP (U-KSH). The authors conducted multiple experiments to illustrate the approximation quality of SHAP-IQ compared to current baseline approaches. They compare the baseline methods with SHAP-IQ using different evaluation metrics. The paper concludes with an appendix that provides further theoretical and experimental results for SHAP-IQ, including additional results for specific CIIs such as SII, n-SII, STI, and FSI. The appendix also includes all proofs of theoretical results from the main paper, further insights into the implementation of SHAP-IQ, and additional experimental results. In summary, this paper presents a novel method for computing Shapley interactions, provides theoretical guarantees for this method, and demonstrates its effectiveness through a series of experiments.

**Strengths:**

**Originality**: The authors propose a novel method, SHAPley Interaction Quantification (SHAP-IQ), for computing Shapley interactions for arbitrary cardinal interaction indices (CII). This is a unique contribution to the field of explainable artificial intelligence (XAI), particularly in the context of black box models. The paper not only presents the new method but also includes a detailed comparison with existing methods, an exploration of its theoretical properties, and a demonstration of its practical application. This comprehensive approach strengthens the authors' claims about the effectiveness of SHAP-IQ.

**Quality**: The paper provides a rigorous theoretical foundation for SHAP-IQ, including proofs of its properties such as being unbiased and consistent. The authors also provide a general approximation bound for SHAP-IQ.

**Clarity**: The paper is well-structured and clearly written. The authors provide a clear explanation of their method and its theoretical underpinnings, making it accessible to readers with a background in the field. The appendix provides additional theoretical and experimental results, further supporting the claims made in the main paper. It also includes all proofs of theoretical results, providing transparency and allowing readers to fully understand the theoretical foundation of SHAP-IQ.

**Significance**: The authors demonstrate the practical application of SHAP-IQ through a series of experiments. They show that it can be applied to different machine learning models and datasets, and that it outperforms existing methods in terms of approximation quality and computational efficiency. This suggests that SHAP-IQ could have a significant impact on the practice of XAI.

**Weaknesses:**

This paper has no obvious weakness. The main concern from me is the reproducibility of the experiments, which seems like requiring too many computational resources.

**Questions:**

Could the authors discuss some future works about the SHAP-IQ?

**Limitations:**

Yes, the authors have discussed limitations well.

---

> ### Author Rebuttal · Authors · 2023-08-08
>
> We would like to sincerely thank the anonymous reviewer for their thoughtful and thorough assessment of our submitted work. We greatly appreciate the detailed list of strengths and contributions of our submitted manuscript.
>
> We strongly believe that the utilization of feature interaction scores will improve the quality of explanations in many application domains.
> Our additional ideas for future work include the following areas:
>
> - **Real-world applications:** Applying SHAP-IQ across different domains, such as NLP and bio-medicine to improve understanding.
> - **Human-centered post-processing:** Post-processing of interaction scores to enhance interpretability for practitioners and ML engineers. For instance, viewing the interaction scores as a local Shapley GAM [5] constitutes an interpretable model that could be further distilled into human-readable code [Ravina.2021].
> -  **Model-specific variants:** Improving the computation of SHAP-IQ, especially in model-specific settings, such as tree-based models, cf. TreeSHAP [24].
> -  **Sequential Evaluation:** Utilize the statistical properties of SHAP-IQ to further improve estimation and generate confidence bounds or conduct hypothesis tests. Possibly, computing SHAP-IQ scores sequentially for tasks such as feature selection or detection of relevant interactions.
>
> **Reference:** [Ravina.2021] Walker Ravina, Ethan Sterling, Olexiy Oryeshko, Nathan Bell, Honglei Zhuang, Xuanhui Wang, Yonghui Wu, Alexander Grushetsky: Distilling Interpretable Models into Human-Readable Code. CoRR abs/2101.08393 (2021)

---

> > ### Comment · Reviewer_xK2B · 2023-08-16
> > **Response to Authors**
> >
> > I am thankful to the authors' response about the future works and I will keep my original score. I suggest authors add these to the camera-ready version if this paper can be accepted.

---

### Official Review · Reviewer_JZ6i · 2023-07-07

**Soundness:** 3 good
**Presentation:** 4 excellent
**Contribution:** 3 good
**Rating:** 5
**Confidence:** 3

**Summary:**

The paper proposes a new sampling-based approximation method called Shapley interaction quantification(SHAP-IQ) for the cardinal interaction index(CII), which is a generalization of the Shapley interaction index. To this end, the authors give a novel representation, and showed that SHAP-IQ is an unbiased and provide a general  approximation bound for the quality and variances of the point estimates. Finally, they demonstrated that SHAP-IQ provides the effectiveness and nice explanations for language model and image classification models.


**Strengths:**

- Provide a unified approach for CII, which is a generalization of unbiased KernelSHAP, and so on.
- SHAP-IQ is an unbiased and its quality and variance are guaranteed to bound theoretically.
- SHAP-IQ empirically provides lowest mean square error compared with other baseline algorithms.

**Weaknesses:**

- The results are solid, but they are not particularly surprising. The contributions are somewhat limited to achieve an unbiased method and theoretically bounding the accuracy.

- Shapley interaction index implies that the number of features to be observed exponentially increases, which in turn may make interpretation more difficult. In some cases, multiple sets of features may have similar importance scores. In such situations, selecting specific sets of features may not be done in a naive way. It would be better to provide a measure to choose them or an approach to reduce the number of feature subsets to be observed.




**Questions:**

- In Figure 2, how are the top-k interactions determined? Are they based on the largest absolute value or some other criterion?
- How should the illustration on the left of Figure 3 be understood?  Each word is assigned both positive and negative values. What does this mean?


**Limitations:**

Yes.

Below are the comments for improvements.
- SOUM is known as the induced subgraph game[1]. The interaction subsets correspond to hyperedges of a graph G and  a is its weight. The value of a subset T is given by the summation of edge weights of a subgraph induced by T.
- In Definition 4.2, K -> k_0? or K seems not be defined.

[1] Deng, Xiaotie, and Christos H. Papadimitriou. 1994. “On the Complexity of Cooperative Solution Concepts.” Mathematics of Operations Research 19 (2): 257–66.

---

> ### Author Rebuttal · Authors · 2023-08-08
>
> We sincerely thank the anonymous reviewer for carefully reading and providing helpful feedback.
>
> - Shapley interaction index implies that the number of features to be observed exponentially increases, which in turn may make interpretation more difficult. In some cases, multiple sets of features may have similar importance scores. In such situations, selecting specific sets of features may not be done in a naive way. It would be better to provide a measure to choose them or an approach to reduce the number of feature subsets to be observed.
>
> Yes, higher order interaction scores need to be filtered or post-processed.
> Human-centered post-processing techniques of feature importance or interaction scores are an interesting and important line of research.
> Yet, in our work, we focused on an efficient and reliable approach to approximate these scores as a foundation.
> The theoretical properties of SHAP-IQ can be used to detect ``significant'' non-zero feature interaction scores sequentially using statistical tools, e.g. hypothesis tests on the estimates, cf. line 209-210.
> In particular, SHAP-IQ allows to selectively estimate interaction scores of interest, in contrast to the baselines of FSI and SII, cf. line 225-226.
>
> - In Figure 2, how are the top-k interactions determined? Are they based on the largest absolute value or some other criterion?
>
> The top-k interactions were determined in regards to absolute value. We will make this clearer by adding a description to line 260-261.
>
> - How should the illustration on the left of Figure 3 be understood? Each word is assigned both positive and negative values. What does this mean?
>
> This visualization of n-SII was introduced in [5], cf. line 284-285.
> For each feature, it displays the sum of positive and negative interaction scores, where this feature is present.
> The interaction scores are thereby distributed equally onto each feature in the feature subset $S$.
> This was theoretically justified in Theorem 6 in [5] and yields that the sum of all bars of each feature are equal to the Shapley value of that feature. We will include an additional line of description of this visualization in line in 284-285.
>
> - SOUM is known as the induced subgraph game [1]. The interaction subsets correspond to hyperedges of a graph G and a is its weight. The value of a subset T is given by the summation of edge weights of a subgraph induced by T.
>
> Indeed, the SOUM could be viewed as an extension of this subgraph game on a hypergraph with edges of different order, i.e. different number of connected nodes.
> We will add this note in the manuscript.
>
> - In Definition 4.2, $K -> k_0$? or K seems not be defined.
>
> $K$ is the number of drawn Monte Carlo samples. We would further clarify this in Definition 4.2.

---

### Official Review · Reviewer_wrD8 · 2023-07-12

**Soundness:** 3 good
**Presentation:** 3 good
**Contribution:** 3 good
**Rating:** 6
**Confidence:** 4

**Summary:**

**Context.** This paper aims at quantifying the contribution of a set of features to the prediction of a machine learning model by leveraging the Shapley value (SV), a principled tool coming from cooperative game theory. Previous works either focused on (i) estimating the contribution of a single feature (via feature importance scores) or on (i) estimating the contribution of a group of features (via feature interaction scores) through different extensions of the SV to features interaction (e.g. SII, STI or FSI). As of today, no unified framework or methodology to compute feature interaction scores has been proposed. This paper tries to fill this gap.

**Contributions.** The authors made the following set of contributions.
* They propose a novel perspective and representation of the cardinal interaction index, agnostic to the size of the features' group to be valued. This representation leads to the SHAP-IQ function unifying previously proposed feature interaction scores.
* They propose a theoretical analysis of SHAP-IQ with both statistic properties (bias, consistency) and non-asymptotic theoretical guarantees in the form of high-probability deviation bounds.
* They illustrate the benefits of their approach on several experiments.


**Strengths:**

* The paper and contributions are timely and of interest for the ML community.
* The paper is clear and well-written.
* I did not fully check in details the Appendix but the first derivations (first 5 pages) in the Appendix are correct and well detailed.
* The related work part spanning both the Introduction and Section 3 is clear and relevant. I appreciated the links made between SHAP-IQ and previous works.
* SHAP-IQ (based on Monte Carlo integration) allows to re-use the same sample for all groups of features.
* Experiments are convincing and well-chosen.

**Weaknesses:**

* Theorem 4.3 is quite classical since SHAP-IQ stands for a Monte Carlo approximation and proofs techniques to obtain high-probability deviation bounds via Chebyshev inequalities have been commonly used in works on Shapley value. Could the authors go beyond this simple theoretical analysis and provide more insights on the impact of key quantities of the problem (e.g. $\nu_0$ and $S$)?
* Full computational complexity and benchmark with competitors: The computational complexity of SHAP-IQ depends on several things: number of Monte Carlo draws, finding the right sampling order $k_0$, pre-computing of the weights $\gamma_s^m$,... Could the authors clarify the full cost and compare it to previous approaches competing with SHAP-IQ?


**Questions:**

See previous section.

**Limitations:**

Yes, done at the end of the paper.

---

> ### Author Rebuttal · Authors · 2023-08-08
>
> We would like to thank the anonymous reviewer for carefully reading and evaluating our submitted manuscript.
>
> - Theorem 4.3 is quite classical since SHAP-IQ stands for a Monte Carlo approximation and proofs techniques to obtain high-probability deviation bounds via Chebyshev inequalities have been commonly used in works on Shapley value. Could the authors go beyond this simple theoretical analysis and provide more insights on the impact of key quantities of the problem (e.g. $\nu_0$ and $S$)?
>
> In our work, we present a novel representation of the CII, which we approximate by a modified Monte Carlo approach (using a sampling order $k_0$).
> We support this method by classical and meaningful theoretical results related to Monte Carlo sampling, such as unbiasedness and consistency. These properties have not been shown for the baselines.
> We further provided a deviation bound that requires as little assumptions as possible.
> In a model-agnostic setting, we suspect that robustness assumptions on $\nu_0$ are required to obtain stronger results, which, in ML context, would imply assumptions on the estimate of the conditional distribution that is used to define $\nu_0$, cf. [3].
> Another promising approach is to restrict the analysis to model-specific games, such as the seminal work on TreeSHAP [24] that computes the Shapley value in polynomial time.
> For tree-based models, we believe that it is possible to exactly compute the CII in polynomial time, which we are interested to study in future work.
>
>
> - Full computational complexity and benchmark with competitors: The computational complexity of SHAP-IQ depends on several things: number of Monte Carlo draws, finding the right sampling order , pre-computing of the weights,...Could the authors clarify the full cost and compare it to previous approaches competing with SHAP-IQ?
>
> For an extensive discussion on the run-time efficiency of the baseline methods we refer to appendix~D.1.3.
>
> 1. **Sampling order $k_0$:** The computation of the sampling order $k_0$ is a constant time operation given a number of features $d$ (it scales linearly with $d$ but this is negligible), cf. Algorithm 2 in the appendix.
> 2. **Weights $\gamma_{s}^{m}$:** The additional computational burden from a pre-computation of the weights $\gamma_{s}^{m}$ is also negligible since it scales linearly with the number of features (number of possible subset sizes) and, in particular does not depend on $\nu$, cf. line 220-221.
> 3. **Main computational costs:** The main computational cost stems from the model evaluations (access to the value function $\nu_{0}(S)$), which is bounded by a model's inference time / scale.
>     We ran a run-time test for the language model (LM) showing how the run-time increases slightly compared to baseline approaches (please see Fig.~1 of the pdf document uploaded under ``Author Rebuttal''). With increasing $K$ the run-time complexity scales linearly, but the overhead of SHAP-IQ is quite low. Note that all pre-computations of SHAP-IQ are included in our evaluation. The difference in STI is a result of less than $K$ model evaluations , cf. lines 15-16 in Algorithm 6 (in the appendix), which is required to maintain efficiency.
>
> The small run-time analysis will be added to the appendix of the manuscript.

---

> > ### Comment · Reviewer_wrD8 · 2023-08-21
> > **Thanks for your answer**
> >
> > I thank the authors for their response and the additional figure provided in the attached .pdf file. I suggest the authors to add these responses in a relevant paragraph in the manuscript to enhance clarity if the latter is accepted. I will keep my score unchanged though since the theoretical analysis is quite classical.

---

### Author Rebuttal · Authors · 2023-08-08

We would like to thank the anonymous reviewers for carefully reading and evaluating our manuscript. As a response to reviewer wrD8, we have uploaded a run-time plot showing the computational effort of SHAP-IQ compared to baseline methods on varying budgets.

---

### Decision · Program_Chairs · 2023-09-21

**Decision:**

Accept (poster)

**Comment:**

This paper looks at evaluating the importance of the interactions of features in a model, an approach sometimes used with the Shapley value technique without the interacting perspective.

It is a good first step toward a nice and formal model of the problem and the reviewers are all positive !